



# Comparing Bayesian and traditional end-member mixing approaches for hydrograph separation in a glacierized basin

Zhihua He[1,#], Katy Unger-Shayesteh[3], Sergiy Vorogushyn[1], Stephan M. Weise[4], Doris Duethmann[5], Olga Kalashnikova[6], Abror Gafurov[1], Bruno Merz[1,2]

[1]GFZ German Research Centre for Geosciences, Section Hydrology, Telegrafenberg, Potsdam, Germany.

[2]University of Potsdam, Institute for Environmental Sciences and Geography, Potsdam, Germany

[3]Now at German Aerospace Center (DLR), International Relations, Linder Höhe, Cologne, Germany

[4]UFZ Helmholtz Centre for Environmental Research UFZ, Department Catchment Hydrology, Halle Germany

[5]Institute of Hydraulic Engineering and Water Resources Management, Vienna University of Technology (TU Wien), Vienna, Austria

[6]CAIAG Central Asian Institute of Applied Geosciences, Department Climate, Water and Natural Resources, Bishkek, Kyrgyzstan

[#]Now at Centre for Hydrology, University of Saskatchewan, Saskatoon, Saskatchewan, Canada





## Abstract

Water tracer data have been successfully used for hydrograph separation in glacierized basins. However, uncertainties in the hydrograph separation are large in these basins, caused by the spatio-temporal variability in the tracer signatures of water sources, the uncertainty of water sampling and the mixing model uncertainty. In this study, we used electrical conductivity (EC) measurements and two isotope signatures ($\delta^{18}O$ and $\delta^2H$) to label the runoff components, including groundwater, snow and glacier meltwater, and rainfall, in a Central Asia glacierized basin. The contributions of runoff components (CRC) to the total runoff, as well as the corresponding uncertainty, were quantified by two mixing approaches: a traditional end-member mixing approach (TEMMA) and a Bayesian end-member mixing approach. The performance of the two mixing approaches were compared in three seasons, distinguished as cold season, snowmelt season and glacier melt season. Results show that: 1) The Bayesian approach generally estimated smaller uncertainty ranges for the CRC compared to the TEMMA. 2) The Bayesian approach tended to be less sensitive to the sampling uncertainties of meltwater than the TEMMA. 3) Ignoring the model uncertainty caused by the isotope fractionation likely leaded to an overestimated rainfall contribution and an underestimated meltwater share in the melt seasons. Our study provides the first comparison of the two end-member mixing approaches for hydrograph separation in glacierized basins, and gives insights for the application of tracer-based mixing approaches for similar basins.



## 1. Introduction

Glaciers and snowpack store a large amount of fresh water in glacierized basins, thus providing an important water source for downstream human societies and ecosystems (Barnett et al., 2005; Viviroli et al., 2007; He et al., 2014; Penna et al., 2016). Seasonal meltwater and rainfall play significant roles in shaping the magnitude and timing of runoff in these basins (Rahman et al., 2015; Pohl et al., 2017). Quantifying the seasonal contributions of the runoff components (CRC), including groundwater, snowmelt, glacier melt and rainfall, to the total runoff is therefore highly needed for the understanding of the dynamics of water resource in glacierized basins under the current climate warming (La Frenierre and Mark, 2014; Penna et al., 2014; He et al., 2015).

The traditional end-member mixing approach (TEMMA) has been widely used for hydrograph separation in glacierized basins across the world (Dahlke et al., 2014; Sun et al., 2016a; Pu et al., 2017). For instance, studies in the Italian glacierized Alpine catchments indicate the successful application of the TEMMA to estimate the proportions of groundwater, snow and glacier meltwater based on water stable isotopes and EC (e.g., Chiogna et al. 2014, Engel et al. 2016 and Penna et al. 2017). Li et al. (2014) confirmed significant contributions of snow and glacier melt runoff to total runoff in the Qilian Mountains using TEMMA. Maurya et al. (2011) reported the contribution of glacial ice meltwater to the total runoff in a Himalayan basin on $\delta^{18}O$ and EC, using a three-component TEMMA.

However, difficulties in field sampling and seasonal inaccessibility often limit the application of TEMMA in high-elevation glacierized basins (Rahman et al., 2015). Moreover, uncertainties for the CRC quantified by the TEMMA are typically high (Klaus and McDonnell, 2013), which can be caused by statistical uncertainty and model uncertainty. Statistical uncertainty refers to the spatio-temporal variability for the tracer signatures, sampling uncertainty and laboratory measurement error (Joerin et al., 2002). Model uncertainty is determined by the assumptions of the TEMMA, which might not agree with reality (Joerin et al., 2002; Klaus and McDonnell, 2013). For example, the fractionation effect on isotope ratios caused by evaporation during the mixing process can result in significant errors given the constant tracer assumption in the TEMMA (Moore and Semmens, 2008).

The Gaussian error propagation technique has been typically applied along with TEMMA to estimate the uncertainty for the hydrograph separation, assuming the uncertainty associated with each source is independent from the uncertainty of other sources (Genereux, 1998; Pu et al., 2013). The spatio-temporal variability for the tracer signatures is estimated by multiplying the *t* values of the Student's *t* distribution at the selected significance level with the


standard deviations (*Sd*) of the measured tracer signatures (Pu et al., 2013; Penna et al., 2016;
Sun et al., 2016b). Although this approach has been successfully used in various glacierized
basins, some recurring issues remain. These include (1) inappropriate estimation of the
variability of tracer signatures of water sources when only few water samples are available
(Dahlke et al., 2014), and (2) negligence of the correlation of water tracers and runoff
components caused by the  assumption of independence of the uncertainty sources. Further, the
model uncertainty caused by the fractionation effect on isotope ratios during the mixing process
is also often ignored.

The Bayesian end-member mixing approach (abbreviated as Bayesian approach) shows

the potential to estimate the proportions of individual components to the mixing variable in a
more rigorous statistical way (Parnell et al., 2010). For hydrograph separation, the water tracer
signatures of the water sources are first assumed to obey specific prior distributions. Their
posterior distribution are then obtained by updating the prior distributions with the observation
likelihood derived from water samples. In the last step, the CRC to the total runoff are estimated
based on the balance of the posterior water tracer signatures. The distributions, expressing the
uncertainties for the CRC and parameters, are typically estimated in a Markov Chain Monte
Carlo (MCMC) procedure.

Although the Bayesian approach can be applied in  cases when the sample sizes are

small (Ward et al., 2010), it has been rarely used for hydrograph separation in glacierized basins.
To the authors' knowledge, there have been only three studies, including Brown et al. (2006),
who conducted the hydrograph separation in a glacierized basin in the French Pyrenees using a
three-component Bayesian approach. Further, Cable et al. (2011) quantified the CRC to total
runoff in a glacierized basin in the American Rocky Mountains. They used a hierarchical
Bayesian framework to incorporate temporal and spatial variability in the water isotope data
into the mixing model. Recently, Beria et al. (2019) used a classic Bayesian approach to
estimate the uncertainty for the CRC in a Swiss alpine catchment. However, the performance
of the Bayesian approach has not been compared to the TEMMA. Moreover, the sensitivity of
the Bayesian approach to the water sampling uncertainty is still not clear. The potential of the
Bayesian approach to estimate the fractionation effect on isotopic signatures during the mixing
process has not been investigated either.

In this study, we compare TEMMA and the Bayesian approach for hydrograph

separation in a Central Asia glacierized basin, using water isotope and EC measurements. The
research questions are: 1) How do TEMMA and the Bayesian approaches compare with respect
to the quantification for the CRC? 2) What is the influence of the different uncertainty sources





(including variability of the tracer signatures, sampling uncertainty, and model uncertainty) on
the estimated CRC in the two mixing approaches?
The paper is organized as follows: details on the study basin and water sampling are
introduced in Section 2; assumptions of the two mixing approaches are described in Section 3;
Section 4 estimates the CRC, as well as the corresponding uncertainties; discussion and
conclusion finalize the paper in Sections 5 and 6, respectively.
**2. Study area and data**
**2.1 Study area**
Located in Kyrgyzstan, Central Asia, the Ala-Archa basin drains an area of 233 km$^2$,
(Fig. 1), and glacier covers around 17% of the basin area. The elevation of the study basin
extends from 1560 m to 4864 m a.s.l.. The seasonal dynamics of runoff in the river play an
important role in the water availability for downstream agricultural irrigation. The generation
of snow and glacier melt runoff generally show the largest effect on the runoff seasonality
(Aizen et al., 2000; Aizen et al., 2007). In particular, the snowmelt runoff mainly occurs in the
warm period from early March to middle September, and the glacier melt typically generates
from the high-elevation areas during July to September (Aizen et al., 1996; He et al., 2018; He
et al., 2019). We subsequently defined three runoff generation seasons as follows. Cold season:
from October to February, in which the streamflow is fed mainly by groundwater and to a
smaller extent by snowmelt and rainfall; Snowmelt season: from March to June, in which the
streamflow is fed chiefly by snowmelt and groundwater and additionally by rainfall; Glacier
melt season: from July to September, in which the streamflow is fed by significant glacier melt
and groundwater, rainfall and snowmelt.
Two meteorological stations (Fig. 1), i.e., Alplager (at elevation of 2100 m a.s.l.) and
Baitik (at elevation of 1580 m a.s.l.), have been set up in the basin since 1960s to collect daily
precipitation and temperature data. The Ala-Archa hydrological station has been set up at the
same site of the Baitik meteorological station to collect daily average discharge data since 1960s.
The dynamics of glacier mass balance and snow mass balance in the accumulation zone have
been surveyed in summer field campaigns through 2012-2017.
**2.2 Water tracer data**
Since July of 2013, stream water samples have been collected weekly by local station
operators, from the river channel close to the Alplager and Baitik meteorological sites, using
pure plastic bottles (He et al., 2019). The sampling time slightly varied around noon every
Wednesday. Precipitation samples were collected during 2012-2017 at four sites across the
basin (Fig. 1). At the Alplager and Baitik meteorological sites, the precipitation samples were


first collected from fixed rain collectors (immediately after the rainfall/snowfall events), and then accumulated in two indoor rain containers over one month. The mixed water in the containers were then sampled for isotopic analysis every month. The indoor rain containers were filled with thin mineral oil layers for monthly precipitation accumulation and stored in cold places. Additionally, two plastic rain collectors PALMEX, specifically designed for isotopic sampling to prevent evaporation, were set up at the elevations of 2580 m a.s.l. and 3300 m a.s.l. to collect precipitation in high-elevation areas (Fig. 1). Precipitation samples were collected monthly from these two rain collectors during the period from May to October when the high-elevation areas were accessible.

Glacier meltwater were sampled during the summer field campaigns in each year of 2012-2017. Samples of meltwater flowing on the Golubin glacier in the ablation zone and at the glacier tongue were collected by pure plastic bottles and then stored in a cooling box (Fig. 1, the elevation of the sampling sites ranges from 3280 m to 3805 m a.s.l.). Snow samples were collected through early March to early October during 2012-2017, as the sampling sites are generally not accessible caused by the heavy snow accumulation in the remaining months. The elevation of the multiple snow sampling sites ranges from 1580 m to 4050 m a.s.l. (Fig. 1). The whole snow profile at each sampling site was collected through drilling a 1.2 m pure plastic tube into the snowpack. The snow in the whole tube were then collected by plastic bags and stored in a cooling box. After all the snow in the plastic bags melted out, the mixed snow meltwater were then sampled by pure plastic bottles. Groundwater samples were also collected through March to October during 2012-2017, from a spring draining to the river (Fig. 1, 2400 m a.s.l.) using pure plastic bottles. The spring is located at the foot of a rocky hill, around 60 meters away from the river channel.

All samples were stored at 4 ℃ and then delivered to the laboratory of Helmholtz Center for Environmental Research (UFZ) in Halle of Germany by flight. Isotopic compositions of water samples were measured using a Laser-based infrared spectrometry (LGR TIWA 45, Picarro L1102-i). The measurement precisions of $\delta^{18}O$ and $\delta^2H$ are: ±0.25 ‰ and ±0.4 ‰, respectively, after the calibration against the common VSMOW standard. EC values of the water samples were measured using portable PH/TDS/EC meters. Abnormal isotopic compositions caused by obvious evaporation and abnormal EC values caused by impurities were discarded.

**3. Methodology**

The hydrograph separation is carried out in each of the three seasons (i.e., clod season, snowmelt season and glacier melt season). Water samples collected in the period from 2012 to




2017 are distributed into each of the three seasons for the hydrograph separation. The CRC
estimated by the mixing approaches refer to the mean contributions in each of the three seasons
during the period of 2012-2017, i.e., the inter-annual variability of CRC were not considered.
The mixing approaches applied for the hydrograph separation in each season are summarized
in Table 2.

**3.1 Traditional end-member mixing approach (TEMMA)**

The main assumptions of TEMMA are as follows (Kong and Pang, 2012): (1) The water
tracer signature of each runoff component is constant during the analyzed period; (2) The water
tracer signatures of the runoff components are significantly different from each other; (3) Water
tracer signatures are conservative in the mixing process. In the cold and snowmelt seasons, a
three-component TEMMA method (TEMMA_3, Table 2) is used. Since the precision of $\delta^{18}O$
(±0.25 ‰) measured in the lab is higher than that of $\delta^2H$ (±0.4 ‰) and both are strongly
correlated, the TEMMA_3 is based on $\delta^{18}O$ and EC. In the glacier melt season, both the
TEMMA_3 and the four-component TEMMA (TEMMA_4, Table 2) are used. In the
TEMMA_3, glacier melt and snowmelt are assumed as one end-member, considering their
similar tracer signatures. In the TEMMA_4, glacier melt and snowmelt are treated as two end-
members separately, and $\delta^{18}O$ and $\delta^2H$ are used as two separate tracers. The following
equations (Eqs. 1-5) are used to estimate CRC ($f_{1-3}$) and the corresponding uncertainty in the
TEMMA_3 (Genereux, 1998).
$$\begin{cases} 1 = f_1 + f_2 + f_3, & \textit{for water balance} \\ A = A_1 \cdot f_1 + A_2 \cdot f_2 + A_3 \cdot f_3, & \textit{for water tracer A} \\ B = B_1 \cdot f_1 + B_2 \cdot f_2 + B_3 \cdot f_3, & \textit{for water tracer B} \end{cases} \quad (1)$$

$$f_1 = \frac{AB_2 - AB_3 + A_2B_3 - A_2B + A_3B - A_3B_2}{A_1B_2 - A_1B_3 + A_2B_3 - A_2B_1 + A_3B_1 - A_3B_2} \quad (2)$$

$$f_2 = \frac{AB_3 - AB_1 + A_1B - A_1B_3 + A_3B_1 - A_3B}{A_1B_2 - A_1B_3 + A_2B_3 - A_2B_1 + A_3B_1 - A_3B_2} \quad (3)$$

$$f_3 = \frac{AB_1 - AB_2 + A_1B_2 - A_1B + A_2B - A_2B_1}{A_1B_2 - A_1B_3 + A_2B_3 - A_2B_1 + A_3B_1 - A_3B_2} \quad (4)$$

where the subscripts 1-3 refer to the three runoff components (i.e., groundwater,
snowmelt/meltwater and rainfall), and $A_1$-$A_3$ ($B_1$-$B_3$) refers to the mean $\delta^{18}O$ (EC) values of
runoff components. $A$ and $B$ stand for the mean $\delta^{18}O$ and EC values of the stream water. The
mean isotope and EC values of precipitation are calculated as the monthly precipitation
weighted average values. Similarly, the mean isotope and EC values of stream water are
calculated as the weekly streamflow weighted average values.





Assuming the uncertainty of each variable is independent from the uncertainty in others,
the Gaussian error propagation technique is applied to estimate the uncertainty of the CRC ($f_{1-}$
$_3$) using the following equation (Genereux, 1998):
$$W_{f_i} = \sqrt{(\frac{\partial f_i}{\partial A_1}W_{A_1})^2 + (\frac{\partial f_i}{\partial A_2}W_{A_2})^2 + (\frac{\partial f_i}{\partial A_3}W_{A_3})^2 + (\frac{\partial f_i}{\partial A}W_A)^2 + (\frac{\partial f_i}{\partial B_1}W_{B_1})^2 + (\frac{\partial f_i}{\partial B_2}W_{B_2})^2 + (\frac{\partial f_i}{\partial B_3}W_{B_3})^2 + (\frac{\partial f_i}{\partial B}W_B)^2}$$
(5)

where $f_i$ stands for the contribution of a specific runoff component, and $W$ is the uncertainty
in the variable specified by the subscript. For the uncertainty of water tracer signatures ($W_{A_i}$
and $W_{B_i}$), we multiply the $Sd$ values of the measured tracer signatures with $t$ values from the
Student's $t$ value table at the confidence level of 95%. The degree of freedom for the
Student's $t$ distribution is estimated as the number of water sample for each water source
minus one. Analytical measurement errors are not considered in this approach, which,
however, are minor compared to the uncertainty generated from water tracer variations
(Penna et al., 2017; Pu et al., 2017). The *lsqnonneg* function in Matlab is used to solve Eqs.
1-4, which solves the equations in a least squares sense, given the constraint that the solution
vector $f$ has nonnegative elements. The TEMMA_4 uses the equations similar to Eqs. 1-5.
**3.2 Bayesian mixing approach**
The Bayesian approaches applied for each season are summarized in Table 2. Similar
to the TEMMA, we apply a three-component Bayesian approach to all seasons, and additionally
a four-component Bayesian approach in the glacier melt season. The three-component Bayesian
approach has two types: the Bayesian_3_Cor approach considers the correlation between $\delta^{18}O$
and $\delta^2H$, whereas the Bayesian_3 approach assumes independence. The four-component
Bayesian approach also has two types: Bayesian_4_Cor considering the correlation, and
Bayesian_4 assuming independence between $\delta^{18}O$ and $\delta^2H$. The prior assumptions for the
Bayesian approaches are listed as follows (similarly to Cable et al. 2011): In approaches
considering the correlation between $\delta^{18}O$ and $\delta^2H$, the prior distributions of $\delta^{18}O$ and $\delta^2H$ of
runoff components and stream water are assumed as bivariate normal distributions with means
and precision matrix as $\mu^{18}O$, $\mu^2H$ and $\boldsymbol{\Omega}$, respectively (Eq.6a). The precision matrix ($\boldsymbol{\Omega}$, i.e. the
inverse of the covariance matrix) for the two isotopes is assumed as Wishart prior (Eq. 6b).
When assuming independence between $\delta^{18}O$ and $\delta^2H$, the prior distributions of $\delta^{18}O$ ($\delta^2H$) of
runoff components and stream water are assumed as normal distributions with means and
variance of $\mu^{18}O$ and $\lambda^{18}O$ ($\mu^2H$ and $\lambda^2H$, Eqs. 6c-d). The mean values of the isotopes of runoff
components (i.e., $\mu^{18}O$ and $\mu^2H$) are further estimated by independent normal priors (Eq. 7,
Cable et al. 2011), which is assumed to consider the spatial variability of $\mu^{18}O$ and $\mu^2H$.



$$\begin{cases} \begin{bmatrix} \delta^{18}O \\ \delta^2H \end{bmatrix} \sim Multi\_normal \left( \begin{bmatrix} \mu^{18}O \\ \mu^2H \end{bmatrix}, \Omega \right) & (6a) \\ \boldsymbol{\Omega} \sim Wishart \ (2, \boldsymbol{V}) & (6b) \\ \delta^{18}O \sim Normal \ (\mu^{18}O, \lambda^{18}O) & (6c) \\ \delta^2H \sim Normal \ (\mu^2H, \lambda^2H) & (6d) \end{cases}$$

$$\begin{cases} \mu^{18}O \sim Normal \ (\gamma^{18}O, \sigma^{18}O) & (7a) \\ \mu^2H \sim Normal \ (\gamma^2H, \sigma^2H) & (7b) \end{cases}$$

where, $\lambda^{18}O$, $\gamma^{18}O$ and $\sigma^{18}O$ ($\lambda^2H$, $\gamma^2H$ and $\sigma^2H$) are parameters used to describe the normal priors of $\delta^{18}O$ and $\mu^{18}O$ ($\delta^2H$ and $\mu^2H$, see Table 3), which are estimated by likelihood observations (Table 3). $\boldsymbol{V}$ is a 2*2 unit positive-definite matrix, and '2' stands for the degree of freedom in the Wishart prior distribution.

The priors of EC values of runoff components and stream water are assumed as normal distributions (Eq. 8a), with mean $\varepsilon$ and variance $\tau$. Similarly, the spatial variability of the mean EC values of runoff components ($\varepsilon$) are assumed to follow a normal distribution with mean $\theta$ and variance $\omega$ (Eq. 8b). $\tau$, $\theta$ and $\omega$ are parameters estimated by likelihood observations (Table 3).

$$\begin{cases} EC \sim Normal \ (\varepsilon, \tau) & (8a) \\ \varepsilon \sim Normal \ (\theta, \omega) & (8b) \end{cases}$$

$$\begin{cases} \begin{bmatrix} \mu^{18}O \\ \mu^2H \\ \varepsilon \end{bmatrix}_{stream \ water} = \sum_{i=1}^{N} f_i \cdot \begin{bmatrix} \mu^{18}O \\ \mu^2H \\ \varepsilon \end{bmatrix}_{runoff \ component \ i} & (9a) \\ \boldsymbol{f} \sim Dirichlet(\boldsymbol{\alpha}) & (9b) \\ \boldsymbol{\alpha} = \boldsymbol{\rho} + \boldsymbol{\psi} & (9c) \\ [\boldsymbol{\rho}, \boldsymbol{\psi}] \sim Multi\_normal(\boldsymbol{\beta}, \boldsymbol{\Omega}) & (9d) \end{cases}$$

The mean isotopes ($\mu^{18}O$ and $\mu^2H$) and EC ($\varepsilon$) of stream water are constrained by a mixing model (Eqs. 9a-b), which estimates the isotope and EC mean values of stream water by multiplying the contribution of each runoff component ($f_i$) with the corresponding mean isotope and EC values of each runoff component (Eq. 9a). In this equation, $N$ is the number of runoff components. The contribution vector ($\boldsymbol{f}$) is represented by a Dirichlet distribution with an index vector $\boldsymbol{\alpha}$ (Eq. 9b), in which the sum of contributions of all runoff components ($\sum f_i$) equals one. The index vector $\boldsymbol{\alpha}$ is estimated by two variable vectors $\boldsymbol{\rho}$ and $\boldsymbol{\psi}$ (Eq.9c), considering the temporal and spatial variability in the CRC (Cable et al. 2011). $\boldsymbol{\rho}$ and $\boldsymbol{\psi}$ are assumed as bivariate





normal distribution with means and precision matrix $\boldsymbol{\beta}$ and $\boldsymbol{\Omega}$ (Eq.9d). $\boldsymbol{\beta}$ is a parameter vector
estimated by likelihood observations (Table 3).
The value ranges for the parameters need to be estimated in Eqs. 6-9 are summarized in
Table 3. The posteriors of parameters describing the spatial variability of water tracers in Eqs.
7 and 8b are first estimated by the mean water tracer signatures of runoff components measured
at different spatial locations. Parameters describing the overall variability of water tracer
signatures in Eqs. 6 and 8a are then constrained by the likelihood observations of water tracer
signatures from all water samples at different times and locations. The posterior distribution of
CRC ($f$) are estimated by Eq. 9, based on the posterior water tracer signatures of runoff
components and the measured water tracer signatures from stream water samples. The
posteriors of parameters and contributions are estimated by the $R$ software package $Rstan$. We
run four parallel Markov Chain Monte Carlo (MCMC) chains with 2000 iterations for each
chain. The first 1000 iterations are discarded for warm-up, generating a total of 4*1000 samples
for the calculation of the posterior distributions. Uncertainties are presented as the 5-95
percentile ranges from the iterative runs. The parameter values are assumed to follow uniform
prior distributions within the value ranges to run the MCMC procedure.
**3.3 Effects of the uncertainty in the meltwater sampling**
Due to limited accessibility, meltwater samples are typically difficult to collect in high-
elevation glacierized areas. Often, only small sample sizes are available to represent the tracer
signatures of meltwater generated from the entire glacierized area. Hence, the
representativeness of meltwater samples can have significant effects on the hydrograph
separation.
To evaluate this effect for the TEMMA and Bayesian mixing approaches, we define
three virtual sampling scenarios. Scenario I: The meltwater sample groups have different
sample sizes, but the same mean value and $Sd$ of the investigated tracer; Scenario II: The
meltwater sample groups have different mean values of the investigated tracer, but the same
sample size and $Sd$ of the investigated tracer; Scenario III: The meltwater sample groups have
different $Sd$ of the investigated tracer, but keeping the same sample size and mean value of the
investigated tracer. We only investigated the effects of the meltwater sampling uncertainty on
the mixing approaches in the glacier melt season, since meltwater is particularly difficult to
collect and is the dominant runoff component in this season. For the water samples of other
runoff components and stream water, we used all the available measurements in the glacier melt
season for the three virtual scenarios, keeping the same sample characteristics.
**3.4 Effects of water isotope fractionation on hydrograph separation**





To consider the changes on the isotope signatures of runoff components caused by the
fractionation effect during the mixing process, we set up two modified Bayesian approaches,
i.e. Bayesian_3_Cor_F and Bayesian_4_Cor_F (Table 2). The effects of water isotope
fractionation on the hydrograph separation are investigated in virtual experiments using the
modified approaches. We modify the mean values in Eq. 9a using fractionation factors $\xi^{18}O$
and $\xi^2H$ (Eq. 10). The priors for $\xi^{18}O$ and $\xi^2H$ are assumed as bivariate normal distributions in
Eq.11.
$$\begin{bmatrix} \mu^{18}O \\ \mu^2H \end{bmatrix}_{stream\ water} = \sum_{i=1}^{N} f_i \bullet \begin{bmatrix} \mu^{18}O + \xi^{18}O \\ \mu^2H + \xi^2H \end{bmatrix}_{runoff\ component\ i} \tag{10}$$

$$\begin{bmatrix} \xi^{18}O \\ \xi^2H \end{bmatrix} \sim Multi\_normal\ (\begin{bmatrix} \eta^{18}O \\ \eta^2H \end{bmatrix}, \boldsymbol{\Omega}) \tag{11}$$

where, $\eta^{18}O$ and $\eta^2H$ are the mean values of the changes in isotopes caused by the fractionation
effect, which are parameters need to be estimated. $\boldsymbol{\Omega}$ is the inverse of the covariance matrix
defined in Eq. 6b. The parameters in Eqs. 6-11 are then re-estimated by the measurements of
water tracer signatures using the MCMC procedure.
**4. Results**
**4.1 Seasonality of water tracer signatures**

Tracer measurements from all the water samples are summarized in Table 1 and Fig. 2.

The mean values indicate that precipitation is most depleted in heavy water isotopes ($^{18}$O and
$^2$H) in the cold season among the water sources. In the melt seasons, snow and glacier meltwater
show the most depleted heavy isotopes. The EC values are highest in groundwater in all seasons,
followed by stream water and precipitation. Snowmelt and glacier melt tend to have the lowest
EC values, due to low interaction with mineral surface.

CV values in Table 1 show that the $\delta^{18}$O and $\delta^2$H of precipitation generally shows the

largest variability in all seasons, followed by the isotopes of snowmelt. Groundwater and stream
water show the smallest CV values for $\delta^{18}$O in all three seasons. The stream water presents the
lowest CV value for EC in all seasons, followed by the groundwater. The snowmelt EC shows
high CV values in the snowmelt and glacier melt seasons, which may be attributed to variable
dust conditions at the sampling locations (from downstream gauge station to upper glacier
accumulation zone). The highest CV value of EC was observed for glacier melt, since the
glacier melt water samples were collected at locations with different sediments conditions in
the ice (from extremely clean to heavily dusty).



329   For each water source except groundwater, the water tracer signatures show a significant

330 seasonality (Table 1). In particular, the $\delta^{18}O$ and $\delta^{2}H$ of precipitation are most depleted in the

331 cold season and reach the highest values in the glacier melt season, partly caused by the

332 seasonality in temperature. Stream water shows higher values of $\delta^{18}O$ and EC in the cold season

333 when groundwater dominates the streamflow, and has lower values in the melt seasons when

334 meltwater has a dominant contribution. Snowmelt has a lower EC value in the glacier melt

335 season than in the cold and snowmelt seasons. This can be explained by the fact that the

336 snowmelt samples in glacier melt season were collected from fresh snow in the accumulation

337 area. The water tracer signature of groundwater is relatively stable across the seasons.

338   Figure 2 shows that the slope of the local meteoric water line (LMWL) is lower than

339 that of the global meteoric water line (GMWL). The $\delta^{18}O$ of precipitation and snowmelt range

340 from -22.82‰ to 1.51‰ and from -17.31‰ to -6.95‰, respectively. The isotopic composition

341 of glacier meltwater is more depleted than those of groundwater and stream water. Stream water

342 shows a similar isotopic composition to groundwater. Three samples from the stream water are

343 far below the LMWL, which is assumed to be caused by the evaporation effect.

344   Figure 3 shows the $\delta^{18}O$-EC mixing space of runoff components in the three seasons.

345 The uncertainty bars of the tracer values represent the temporal and spatial variability. In the

346 cold season, the $\delta^{18}O$ and EC values of stream water are very close to those of groundwater

347 (Fig. 3a), whereas the snowmelt and precipitation tracer signatures are different. These results

348 indicate the dominance of groundwater on streamflow during the cold season. In the snowmelt

349 and glacier melt seasons (Figs. 3b-c), the stream water samples are located clearly within the

350 triangle formed by the samples of runoff components. The water tracer signatures of glacier

351 meltwater and snowmelt water are similar. The precipitation samples are farther away from the

352 stream water samples compared to the meltwater and groundwater samples. The stream water

353 samples are located nearly in the middle between the meltwater and groundwater samples. This

354 indicates that the contribution of rainfall to total runoff is smallest and the contributions of

355 meltwater and groundwater are similar, in the melt seasons. We assume the tracer signatures of

356 rainfall are represented by the measurements of precipitation samples in all three seasons.

357 **4.2 Contributions of runoff components estimated by the mixing approaches**

358   Table 4 and Fig. 4 compare the CRC estimated by multiple mixing approaches. In the

359 cold season (Fig. 4a), the TEMMA_3 estimated the mean contributions of groundwater and

360 snowmelt as 83% and 17%, respectively. The mean contribution of rainfall is zero. The mean

361 contributions of groundwater, snowmelt and rainfall were estimated as 86% (87%), 13% (12%)

362 and 1% (1%) by the Bayesian_3 (Bayesian_3_Cor) approach. As shown in Fig. 3a, the water

tracer signature of stream water in this season is close to that of groundwater, while obviously
different from that of rainfall. Meanwhile, the stream water samples are outside of the triangle
formed by the runoff components, leading to the zero contribution of the rainfall estimated by
the TEMMA_3. The ranges for the CRC indicate the uncertainty in the estimates associated
with the corresponding mixing approaches (Table 4). The TEMMA_3 produced the highest
uncertainty for the CRC, followed by the Bayesian_3. The Bayesian_3_Cor slightly reduced
the uncertainty compared to the Bayesian_3, benefiting from the consideration of the
correlation between $\delta^{18}$O and $\delta^2$H.

In the snowmelt season (Fig. 4b and Table 4), the TEMMA_3 estimated the mean

contributions of groundwater, rainfall and snowmelt as 44%, 36% and 20%, respectively. The
Bayesian_3 estimated similar mean CRC to the TEMMA_3, whereas the Bayesian_3_Cor
delivered a lower contribution of snowmelt (32%). When treating the glacier melt and snowmelt
as one end-member (i.e. meltwater) in the glacier melt season (Fig. 4c), the TEMMA_3
estimated the mean contributions of groundwater, meltwater and rainfall of 45%, 46% and 9%,
respectively. The Bayesian_3 and Bayesian_3_Cor estimated a lower contribution of
groundwater (43-44%) and a higher contribution of rainfall (11%) compared to the TEMMA_3.
In general, the TEMMA_3 estimated the largest uncertainty for the contributions in all the three
seasons, followed by the Bayesian_3. The Bayesian_3_Cor slightly reduced the uncertainty
ranges compared to the Bayesian_3 (Table 4).

When treating glacier melt and snowmelt as two separate end-members in the glacier

melt seasons (Fig. 4d), the TEMMA_4 failed to separate the hydrograph in the glacier melt
season, given the large uncertainty range for the contributions of snowmelt and rainfall (0-
100%). The tracer signatures of snow and glacier meltwater are rather close to each other, that
violates the second assumption of the TEMMA (see Sec. 3.1). In contrast, the Bayesian_4_Cor
and Bayesian_4 estimated the shares of glacier melt and snowmelt as 25-24% and 21-25%,
respectively. Considering the significant snow cover area in September in the study basin (He
et al. 2018; He et al. 2019), the contribution of snowmelt in the glacier melt season should be
much higher than zero. Again, the Bayesian_4_Cor produced smaller uncertainty ranges for the
contributions of groundwater and meltwater compared to the Bayesian_4 and TEMMA_4
(Table 4).

The posterior distributions of water tracer signatures estimated by the Bayesian_4_Cor

in the glacier melt season are compared with the measured distributions of water tracers in Fig.
5. The Bayesian_4_Cor generally produced similar distributions of water isotopes to the
measured distributions, in terms of the similar mean values. The estimated posterior *Sd* values




of the water isotopes are smaller than those of the measured water isotopes. This can be
explained by the incorporation of prior distributions by the Bayesian_4_Cor, thus reducing the
variability of water isotopes. The posterior *Sd* values for the EC of water sources are also
smaller than the measured *Sd* values. However, the posterior distributions of EC show some
deviations from the distributions of measured EC, partly due to the very small sample sizes (see
Table 1). The comparison between the posterior distributions of water tracers estimated by the
Bayesian_3_Cor and the measured distributions in the other seasons generally shows a similar
behavior (not shown for brevity).
The Bayesian_4 estimated similar posterior distributions of water tracer signatures to
the Bayesian_4_Cor (except the glacier melt isotopes, Fig. 6), with similar mean tracer values
and *Sd*. It is noted that the Bayesian_4_Cor estimated smaller *Sd* values for most water sources
than the Bayesian_4 (e.g., Figs. 6f-g and 6i-j). Benefiting from the prior information and the
consideration of the correlation between $\delta^{18}O$ and $\delta^2H$, the Bayesian_4_Cor tended to produce
the smallest variability in the posterior water tracers among the mixing approaches (Figs. 5-6),
thus resulting in the smallest uncertainty for CRC (Fig. 4d). Figure 7 compares the correlation
between $\delta^{18}O$ and $\delta^2H$ in the measured tracers and the posterior estimates by the Bayesian
approaches. The Bayesian_4_Cor reproduced the correlation between $\delta^{18}O$ and $\delta^2H$ well in
comparison to the measured data, whereas the Bayesian_4 failed to capture the correlation.

### 4.3 Uncertainty for hydrograph separation caused by sampling uncertainty of meltwater

Figure 8 shows the sensitivity of the Bayesian_3_Cor and TEMMA_3 approaches to the
sampled $\delta^{18}O$ of meltwater in the glacier melt season. The mean CRC quantified by the two
mixing approaches show minor sensitivity to the sample size (scenario I). However, the
uncertainty ranges for the contributions tend to decrease with increasing sample size, especially
for the TEMMA_3. When assuming only two meltwater samples, the TEMMA_3 resulted in
very large uncertainty ranges (0-100%), due to the very wide confidence interval for the *Sd* at
a sample size of two. The mean contributions of groundwater and meltwater estimated by the
two mixing approaches decrease with increasing mean $\delta^{18}O$ of the adopted meltwater sample
(scenario II), while the estimated contribution of rainfall increases with the increasing mean
$\delta^{18}O$. The variations in the mean CRC quantified by the TEMMA_3 are larger than those
estimated by the Bayesian_3_Cor. In the TEMMA_3, both the mean contributions of
groundwater and meltwater declined by 9% with the assumed increase of the mean $\delta^{18}O$, and
the contribution of rainfall increased by 17%. In the Bayesian_3_Cor, the reduction for the
contributions of groundwater and snowmelt are 4% and 7%, respectively, and the increase for
the contribution of rainfall is 11%. In scenario III, the uncertainty ranges for the CRC
(especially for rainfall, Fig. 8l) increase with increasing $Sd$ of the sampled $\delta^{18}$O. Again, the
increases in the uncertainty ranges estimated by the TEMMA_3 tend to be larger than those
estimated by the Bayesian_3_Cor. The sensitivity of the mixing approaches to the sampled EC
values of the meltwater are similar to the sensitivity to the sampled $\delta^{18}$O (not shown).

**4.4 Effect of isotope fractionation on the hydrograph separation**

The changes of $\delta^{18}$O caused by the fractionation effect during the mixing process are
estimated in Figs. 9a-c. The fractionation has the smallest effect on the $\delta^{18}$O of groundwater,
while the largest effect on the $\delta^{18}$O of rainfall. Averagely, the $\delta^{18}$O of rainfall was increased by
around 2.8‰ through the fractionation. The CRC estimated by the Bayesian_3_Cor_F and
Bayesian_4_Cor_F are compared with those estimated by the Bayesian_3_Cor and
Bayesian_4_Cor in Figs. 9d-f, respectively. The mean contribution of groundwater estimated
by the Bayesian_3_Cor_F in the cold season is 9% lower than that estimated by the
Bayesian_3_Cor (Fig. 9d), while the mean contributions of snowmelt and rainfall are 3% and
5% higher, respectively. The reduction of groundwater contribution is the compensation for the
increased contributions of snowmelt and rainfall caused by the fractionation effect. In the
snowmelt season, the mean contributions of groundwater and rainfall are 1% and 7% lower
(Fig. 9e), while the mean contribution of snowmelt estimated by the Bayesian_3_Cor_F is 8%
higher. In the glacier melt season, the mean contributions of groundwater and meltwater
estimated by the Bayesian_4_Cor_F are higher than those estimated by the Bayesian_4_Cor
(Fig. 9f) and are compensated by the 6% lower contribution of rainfall.
The fractionation effect also produced visible changes on the posterior distributions of
$\delta^{18}$O and $\delta^{2}$H of runoff components (Fig. 10 shows the example in the glacier melt season). The
mean isotopic compositions of runoff components are increased by the fractionation effect. The
$Sd$ values of the posterior isotopes estimated by the Bayesian_4_Cor_F tend to be higher than
those estimated by the Bayesian_4_Cor, due to the increased parameter space in the prior
assumptions (Eq. 11), thus leading to the larger uncertainty ranges for the contributions of
glacier melt and snowmelt (Fig. 9f). As expected, the estimates for the posterior distributions
of isotopic compositions of stream water are less sensitive to the fractionation effect of runoff
components (Figs. 10e and 10j). The fractionation also has minor effects on the estimates for
the posterior distributions of EC values (Figs. 10k-o).

**5. Discussion**

**5.1 Uncertainty for the contributions of runoff components**

The TEMMA estimated larger uncertainties for the CRC in comparison to the Bayesian
approaches. The reasons for this are two-fold. First, the TEMMA estimated the uncertainty





ranges for the CRC using the standard deviations (*Sd*) of the measured water tracer signatures.
*Sd* is likely overestimated, due to small sample size and thus insufficiently represents the
variability of the tracers of the corresponding water sources. Due to the limited accessibility of
the sampled sites caused by snow cover, the water samples of meltwater and groundwater are
often collected occasionally, thus leading to sharp changes in the measured water tracer
signatures. Second, the TEMMA assumes that the uncertainty associated with each water source
is independent from the uncertainty of other water sources (Eq.5), which increases the
uncertainty ranges for CRC.

In contrast, the Bayesian approaches estimated smaller variability of water tracer

signatures in the posterior distributions compared to the measured water tracer signatures, by
updating the prior probability distributions. The posterior distributions were sampled
continuously from the assumed value ranges, thus reducing the sharp changes and yielding
lower variability for the tracer signatures. Moreover, the uncertainty ranges for CRC were
quantified using Eqs. 6-10, instead of calculating independently as in the TEMMA.
Additionally, the assumed prior distributions for the water tracers and the CRC take into
account the correlation between the water tracers and the dependence between the runoff
components in the Bayesian approaches, thus resulting in smaller uncertainty ranges (Soulsby
et al., 2003). For example, the Bayesian approaches considering the correlation between $\delta^{18}O$
and $\delta^2H$ generally estimated smaller uncertainty ranges for CRC compared to those without
considering this correlation.

The Gaussian error propagation technique is only capable of considering the uncertainty

for the CRC resulting from the variation in the water tracer signatures (Uhlenbrook and Hoeg,
2003). The uncertainty for CRC originated from the sampling uncertainty of meltwater was
then investigated in separate virtual sampling experiments. The TEMMA produces large
uncertainty ranges in the glacier melt season, when the meltwater sample size is rather small.
The mean CRC quantified by the TEMMA rely more heavily on the mean tracer values of the
sampled meltwater, as the mean tracer values are directly used in Eqs. 1-4, in comparison to the
mean CRC estimated by the Bayesian approach.

The TEMMA assumes that the water tracer signature of each runoff component is

constant during the mixing process, thus is unable to estimate the uncertainty for CRC caused
by the isotope fractionation effect. The virtual fractionation experiments using the modified
Bayesian approaches show that the isotope fractionation could increase the contribution of
snowmelt by 8%, and reduce the contribution of rainfall by 7% in the snowmelt season. We
assume the mean CRC estimated by the Bayesian approaches considering the isotope





fractionation are more plausible, though the larger uncertainty ranges. Along the flow path from
the source areas to river, the isotopic compositions of meltwater and rainfall are likely increased
by the evaporation fractionation effect, especially in the warm seasons. The increased isotopic
compositions of meltwater and rainfall during the routing process need to be considered in the
mixing approaches for hydrograph separation.

In general, the uncertainty for the CRC is visibly caused by the spatio-temporal

variability in the water tracer signatures, the water sampling uncertainty and the isotope
fractionation during the mixing process. The uncertainty caused by the water sampling of
meltwater tends to be smaller than the uncertainty caused by the variations of the water tracer
signatures in both the TEMMA and Bayesian mixing approaches. This is consistent to the
findings that the *Sd* values in the tracer measurements of water samples are the main uncertainty
sources for the CRC (Schmieder et al., 2016; Schmieder et al., 2018). The Bayesian approach
tends to be superior in narrowing the variability of posterior water tracer signatures benefiting
from the prior assumptions and the consideration of the dependence between water tracer
signatures and runoff components compared to the TEMMA.

**5.2 Limitations**

The representativeness of the water samples is one of the limitations of this study. The

groundwater was only sampled from a single spring located at the elevation of 2400 m a.s.l,
which is rather close to the average altitude of the entire river network in the study basin (2530
m a.s.l.). We thus assume that the measured isotopic composition of the spring water represents
the mean isotopic composition of groundwater feeding the river in the basin (similarly to He et
al., 2019). Collecting samples from a few spring points to represent the groundwater end-
member has been proposed before (such as Ohlanders et al., 2013 and Mark and McKenzie,
2007), as the accessibility and availability of more potential springs are hampered. Again, for
the snow and glacier meltwater samples, we assume that meltwater occurring at similar
elevations have similar water tracer signatures (He et al., 2019). The sampled elevation ranges
from 1580 m to 4050 m a.s.l., matching with the elevation range where meltwater mainly occurs
in the basin (from 1580 m to 3950 m a.s.l.). The sampled sites thus bear the potential to provide
the water tracer signatures for the major share of the meltwater generated in the basin. We
divided the entire sampling period (years of 2012 to 2017) into three seasons, i.e. cold season,
snowmelt season and glacier melt season, due to the low availability of water samples in each
year. By concentrating water samples in the three seasons, we increased the sample sizes of
each runoff component for each season, thus increasing the ability of water samples to represent
the spatio-temporal variability of seasonal tracer signatures.





533   The assumptions of the mixing approaches lead to another limitation of this study. The

534   TEMMA assumes the tracer signatures of water sources are constant during the mixing process,

535   which is a common assumption for TEMMA. It thus fails to consider the uncertainty originating

536   from the changes of water tracers. In the Bayesian approach, we assumed normal prior

537   distributions for the water tracers of water sources and Dirichlet prior distribution for the CRC

538   by literature knowledge (Cable et al., 2011). To refine the description of the temporal and spatial

539   variability of the CRC in the Dirichlet distribution, more hydrological data relating to the runoff

540   processes in the basin are required. We acknowledge that the estimated CRC could be strongly

541   affected by the assumptions of prior distributions. However, testing the effects of the prior

542   assumptions goes beyond the scope of this study. We assume that collecting more water

543   samples from various locations and at different time for each water source could improve the

544   estimation for the tracer signature distributions.

**6. Conclusions**

546   This study compared the Bayesian end-member mixing approach with a traditional end-

547   member mixing approach (TEMMA) for hydrograph separation in a glacierized basin. The

548   contributions of runoff components (CRC) to the total runoff were estimated for three seasons,

549   i.e. cold season, snowmelt and glacier melt seasons. Uncertainty for these contributions caused

550   by the variability of water tracer signatures, water sampling uncertainty and isotope

551   fractionation were evaluated as follows.

552   (1) The Bayesian approach generally estimates smaller uncertainty ranges for the CRC,

553   in comparison to the TEMMA. Benefiting from the prior assumptions on water tracer signatures

554   and CRC, as well as from the incorporation of the correlation between tracer signatures in the

555   prior distributions, the Bayesian approach reduced the uncertainty. The Bayesian approach

556   jointly quantified the uncertainty ranges for the CRC. In contrast, the TEMMA estimated the

557   uncertainty for the contribution of each runoff component independently, thus leading to higher

558   uncertainty ranges.

559   (2) The estimates for CRC in the TEMMA tend to be more sensitive to the sampling

560   uncertainty of meltwater, compared to those in the Bayesian approach. For small sample sizes

561   (e.g., two), the TEMMA estimated very large uncertainty ranges. The mean CRC quantified by

562   the TEMMA are also more sensitive to the mean value of the tracer signature of the meltwater

563   samples than those estimated by the Bayesian approach are.

564   (3) Ignoring the isotope fractionation during the mixing process likely overestimates the

565   contribution of rainfall and underestimates the contribution of meltwater in the melt seasons.



The currently used TEMMA is unable to quantify the uncertainty for CRC caused by the isotope
fractionation during the mixing process, due to the underlying assumptions.



Code availability: The R code for the Bayesian end-member mixing approach can be found at
https://www.dropbox.com/s/kf2xy3s4vt718s9/Bayesian%20mixing%20approach_four%20co
mponents.stan?dl=0

Author contributions.
Conceptualization: Zhihua He, Katy Unger-Shayesteh, and Sergiy Vorogushyn; Data collection:
Zhihua He, Katy Unger-Shayesteh, Stephan M. Weise, Olga Kalashnikova, and Abror Gafurov;
Methodology: Zhihua He, Katy Unger-Shayesteh, and Sergiy Vorogushyn; Writing original
draft: Zhihua He, Sergiy Vorogushyn, and Doris Duethmann: Writing review and editing, All

Competing interests.
The authors declare no conflict of interest.

**Acknowledgement**
Our work has been funded by the German Federal Ministry for Science and Education (project
GlaSCA-V, grant number 88 501) and Volkswagen Foundation (project GlaSCA, grant number
01DK15002A and B), respectively.





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





**LIST OF TABLES**





Table 1. Water tracer signatures measured from water samples in three seasons. CV is the
ratio between the standard deviation and mean value.

| Season | Water source | Tracer | Sample size | Mean | Range | CV |
|---|---|---|---|---|---|---|
| | Groundwater | $^{18}O$ (δ,‰) | 23 | -11.37 | (-12.12, -10.61) | 0.04 |
| | | $^2H$ (δ,‰) | 23 | -73.9 | (-77.9, -68.2) | 0.03 |
| | | EC(μs/cm) | 13 | 126.8 | (69.6, 167.2) | 0.24 |
| | Precipitation | $^{18}O$ (δ,‰) | 37 | -15.93 | (-22.82, -7.70) | 0.21 |
| | | $^2H$ (δ,‰) | 37 | -111.5 | (-168.8, -39.1) | 0.27 |
| | | EC(μs/cm) | 23 | 67.8 | (21.3, 99.6) | 0.34 |
| Cold season | Snowmelt | $^{18}O$ (δ,‰) | 36 | -12.51 | (-17.31, -6.95) | 0.19 |
| (October to February) | | $^2H$ (δ,‰) | 36 | -84.6 | (-120.7, -38.7) | 0.23 |
| | | EC(μs/cm) | 15 | 53.7 | (8.8, 151) | 0.96 |
| | Stream water | $^{18}O$ (δ,‰) | 150 | -11.33 | (-11.82, -9.05) | 0.03 |
| | | $^2H$ (δ,‰) | 150 | -74.2 | (-77.5, -68.2) | 0.03 |
| | | EC(μs/cm) | 90 | 112.2 | (80.3, 139.3) | 0.13 |
| | Groundwater | $^{18}O$ (δ,‰) | 9 | -11.34 | (-11.94, -11.06) | 0.02 |
| | | $^2H$ (δ,‰) | 9 | -73.9 | (-77.3, -72.4) | 0.02 |
| | | EC(μs/cm) | 8 | 133.1 | (94, 167.2) | 0.21 |
| | Precipitation | $^{18}O$ (δ,‰) | 25 | -7.89 | (-16.81, -0.06) | 0.46 |
| | | $^2H$ (δ,‰) | 25 | -49.2 | (-120.5, -3.9) | 0.52 |
| | | EC(μs/cm) | 11 | 58.3 | (25.8, 84.3) | 0.34 |
| Snowmelt season | Snowmelt | $^{18}O$ (δ,‰) | 15 | -13.87 | (-16.74, -10.96) | 0.11 |
| (March to June) | | $^2H$ (δ,‰) | 15 | -95.9 | (-119.3, -70.5) | 0.13 |
| | | EC(μs/cm) | 11 | 67.3 | (11.0, 151.0) | 0.80 |
| | Stream water | $^{18}O$ (δ,‰) | 126 | -11.58 | (-12.91, -10.04) | 0.04 |
| | | $^2H$ (δ,‰) | 126 | -76.1 | (-86.4, -67.0) | 0.04 |
| | | EC(μs/cm) | 23 | 94.9 | (80.1, 114.0) | 0.09 |
| | Groundwater | $^{18}O$ (δ,‰) | 14 | -11.4 | (-12.12, -10.61) | 0.04 |
| | | $^2H$ (δ,‰) | 14 | -73.9 | (-77.9, -68.2) | 0.04 |
| | | EC(μs/cm) | 5 | 116.7 | (69.6, 142.6) | 0.30 |
| | Precipitation | $^{18}O$ (δ,‰) | 28 | -6.72 | (-13.02, 1.51) | 0.56 |
| | | $^2H$ (δ,‰) | 28 | -42.6 | (-94.9, 3.0) | 0.58 |
| | | EC(μs/cm) | 9 | 67.7 | (26.7, 102.0) | 0.39 |
| Glacier melt season | Snowmelt | $^{18}O$ (δ,‰) | 15 | -12.70 | (-17.31, -9.85) | 0.15 |
| (July to September) | | $^2H$ (δ,‰) | 15 | -85.6 | (-120.7, -64.0) | 0.17 |
| | | EC(μs/cm) | 4 | 16.2 | (8.8, 24.3) | 0.51 |
| | Glacier melt | $^{18}O$ (δ,‰) | 23 | -13.11 | (-14.96, -11.55) | 0.10 |
| | | $^2H$ (δ,‰) | 23 | -87.2 | (-100.4, -75.5) | 0.11 |
| | | EC(μs/cm) | 10 | 9.9 | (1.5, 33.4) | 1.28 |
| | Stream water | $^{18}O$ (δ,‰) | 119 | -11.75 | (-12.97, -5.64) | 0.07 |
| | | $^2H$ (δ,‰) | 119 | -77.2 | (-86.7, -62.3) | 0.05 |
| | | EC(μs/cm) | 24 | 64.5 | (33.4, 99.3) | 0.25 |






Table 2. Mixing approaches used for hydrograph separation in different seasons.

| Mixing approach | Description | End-member | Used tracers | Seasons applied to |
|---|---|---|---|---|
| TEMMA_3 | Three-component traditional end-member mixing approach | Groundwater, snowmelt (or meltwater) and rainfall | $^{18}$O and EC | Cold season, snowmelt season and glacier melt season |
| TEMMA_4 | Four-component traditional end-member mixing approach | Groundwater, snowmelt, glacier melt and rainfall | $^{18}$O, $^{2}$H and EC | Glacier melt season |
| Bayesian_3 | Three-component Bayesian approach, without considering the correlation between $\delta^{18}$O and $\delta^{2}$H | Groundwater, snowmelt (or meltwater) and rainfall | $^{18}$O and EC | Cold season, snowmelt season and glacier melt season |
| Bayesian_3_Cor | Three-component Bayesian approach, considering the correlation between $\delta^{18}$O and $\delta^{2}$H | Groundwater, snowmelt (or meltwater) and rainfall | $^{18}$O, $^{2}$H and EC | Cold season, snowmelt season and glacier melt season |
| Bayesian_3_Cor_F | Three-component Bayesian approach, considering the correlation between $\delta^{18}$O and $\delta^{2}$H and the fractionation of $\delta^{18}$O and $\delta^{2}$H during the mixing process | Groundwater, snowmelt and rainfall | $^{18}$O, $^{2}$H and EC | Cold season and snowmelt season |
| Bayesian_4 | Four-component Bayesian approach, without considering the correlation between $^{18}$O and $^{2}$H | Groundwater, snowmelt, glacier melt and rainfall | $^{18}$O, $^{2}$H and EC | Glacier melt season |
| Bayesian_4_Cor | Four-component Bayesian approach, considering the correlation between $\delta^{18}$O and $\delta^{2}$H | Groundwater, snowmelt, glacier melt and rainfall | $^{18}$O, $^{2}$H and EC | Glacier melt season |
| Bayesian_4_Cor_F | Four-component Bayesian approach, considering the correlation between $\delta^{18}$O and $\delta^{2}$H and the fractionation of $\delta^{18}$O and $\delta^{2}$H during the mixing process | Groundwater, snowmelt, glacier melt and rainfall | $^{18}$O, $^{2}$H and EC | Glacier melt season |




Table 3. Parameters used for the prior distributions in the Bayesian approaches.

| Parameter | Description | Applied Bayesian approach | Value range | Equation |
|---|---|---|---|---|
| $\gamma^{18}O$ | Mean of the prior normal distributions for the mean $\delta^{18}O$ of runoff components | All Bayesian approaches | (-50,50) | Eq.7a |
| $\gamma^{2}H$ | Mean of the prior normal distributions for the mean $\delta^{2}H$ of runoff components | All Bayesian approaches, except Bayesian_3 | (-200,200) | Eq.7b |
| $\sigma^{18}O$ | Variance of the prior normal distributions for the mean $\delta^{18}O$ of runoff components | All Bayesian approaches | (0,50) | Eq.7a |
| $\sigma^{2}H$ | Variance of the prior normal distributions for the mean $\delta^{2}H$ of runoff components | All Bayesian approaches, except Bayesian_3 | (0,200) | Eq.7b |
| $\lambda^{18}O$ | Variance of the prior normal distributions for the $\delta^{18}O$ of runoff components and stream water | Bayesian_3 and Bayesian_4 | (0,50) | Eq.6c |
| $\lambda^{2}H$ | Variance of the prior normal distributions for the $\delta^{2}H$ of runoff components and stream water | Bayesian_4 | (0,200) | Eq.6d |
| $\tau$ | Variance of the prior normal distributions for the EC of runoff components and stream water | All Bayesian approaches | (0,400) | Eq.8a |
| $\theta$ | Mean of the prior normal distributions for the mean EC of runoff components | All Bayesian approaches | (0,400) | Eq.8b |
| $\omega$ | Variance of the prior normal distributions for the mean EC of runoff components | All Bayesian approaches | (0,400) | Eq.8b |
| $\beta$ | Mean of the prior bivariate normal distributions for parameters describing the $\alpha$ value in the Dirichlet distribution of contributions of runoff components | All Bayesian approaches | (0,10) | Eq.9d |
| $\eta^{18}O$ | Mean of the prior bivariate normal distributions for the fractionations of $\delta^{18}O$ of runoff components | Bayesian_3_Cor_F and Bayesian_4_Cor_F | (0,5) | Eq.11 |
| $\eta^{2}H$ | Mean of the prior bivariate normal distributions for the fractionations of $\delta^{2}H$ of runoff components | Bayesian_3_Cor_F and Bayesian_4_Cor_F | (0,5) | Eq.11 |






Table 4. Contributions of runoff components (CRC) estimated by the different mixing
approaches (%). The ranges show the difference between the 95% and 5% percentiles.

|  | Mixing approach | Groundwater | | Snowmelt | | Rainfall | | Glacier melt | | Meltwater | |
|---|---|---|---|---|---|---|---|---|---|---|---|
|  |  | Mean | Range | Mean | Range | Mean | Range | Mean | Range | Mean | Range |
| Cold season | TEMMA_3 | 83 | 41 | 17 | 46 | 0 | 10 | - | - | - | - |
|  | Bayesian_3 | 86 | 28 | 13 | 28 | 1 | 3 | - | - | - | - |
|  | Bayesian_3_Cor | 87 | 24 | 12 | 24 | 1 | 3 | - | - | - | - |
| Snowmlet season | TEMMA_3 | 44 | 50 | 36 | 33 | 20 | 25 | - | - | - | - |
|  | Bayesian_3 | 42 | 33 | 36 | 22 | 22 | 20 | - | - | - | - |
|  | Bayesian_3_Cor | 46 | 30 | 32 | 20 | 22 | 19 | - | - | - | - |
| Glacier melt season (three-component) | TEMMA_3 | 45 | 48 | - | - | 9 | 17 | - | - | 46 | 35 |
|  | Bayesian_3 | 43 | 25 | - | - | 11 | 13 | - | - | 46 | 18 |
|  | Bayesian_3_Cor | 44 | 24 | - | - | 11 | 12 | - | - | 45 | 17 |
| Glacier melt season (four-component) | TEMMA_4 | 45 | 48 | 0 | 100 | 11 | 100 | 44 | 78 | - | - |
|  | Bayesian_4 | 44 | 30 | 21 | 42 | 10 | 13 | 25 | 41 | - | - |
|  | Bayesian_4_Cor | 41 | 23 | 25 | 33 | 10 | 13 | 24 | 33 | - | - |






**LIST OF FIGURES**



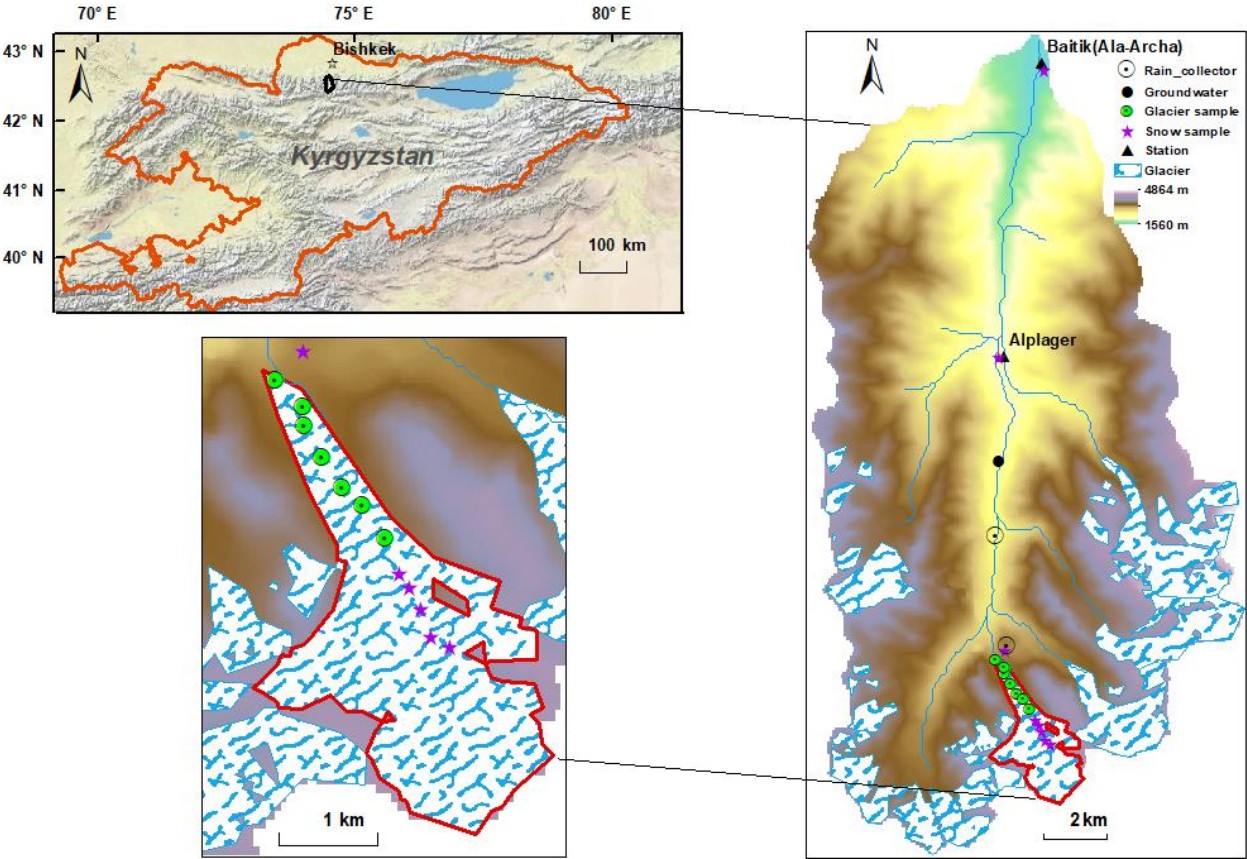

Figure 1. Study area of the Ala-Archa basin (derived from the ESRI World Topographic Map)
and the Golubin Glacier including the locations of the water sampling points.

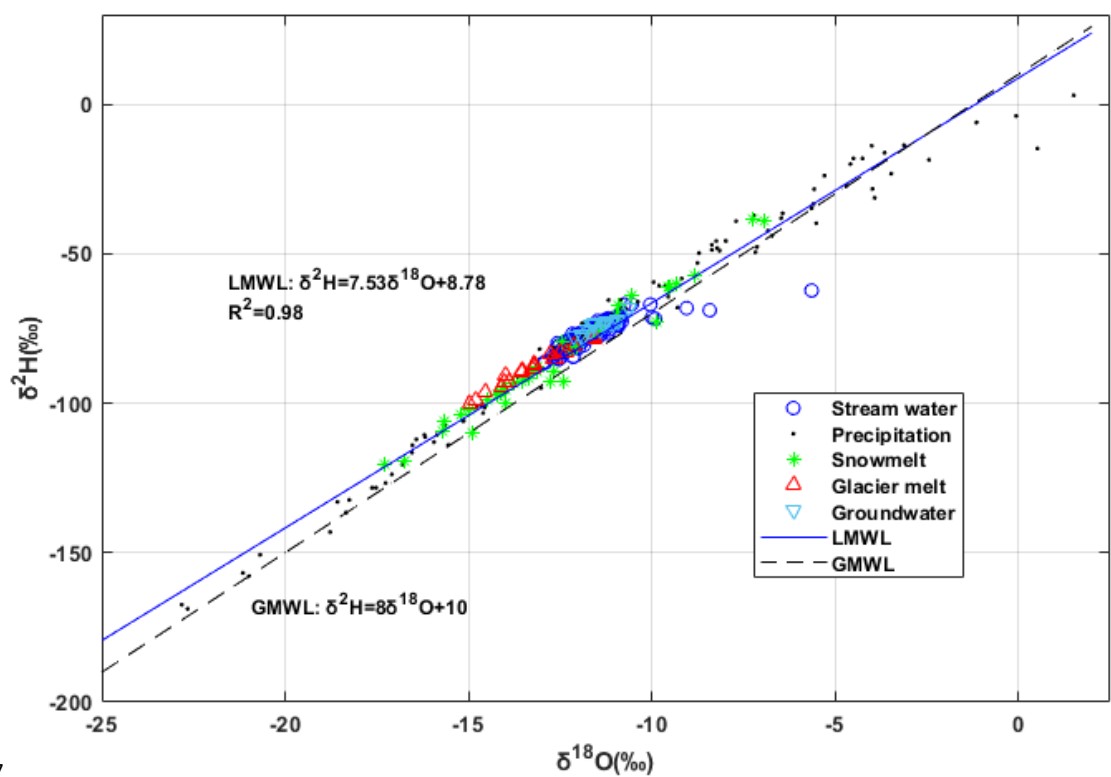

Figure 2. Isotope signatures of water samples from the three seasons in the Ala-Archa basin.





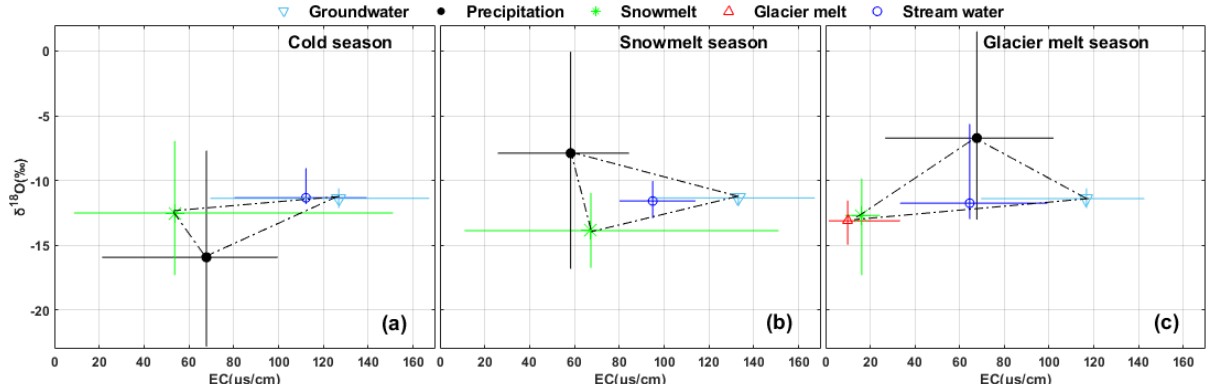

Figure 3. $\delta^{18}$O-EC mixing space of the various water sources in the three seasons. The solid

lines indicate the ranges of tracer signatures measured from water samples.



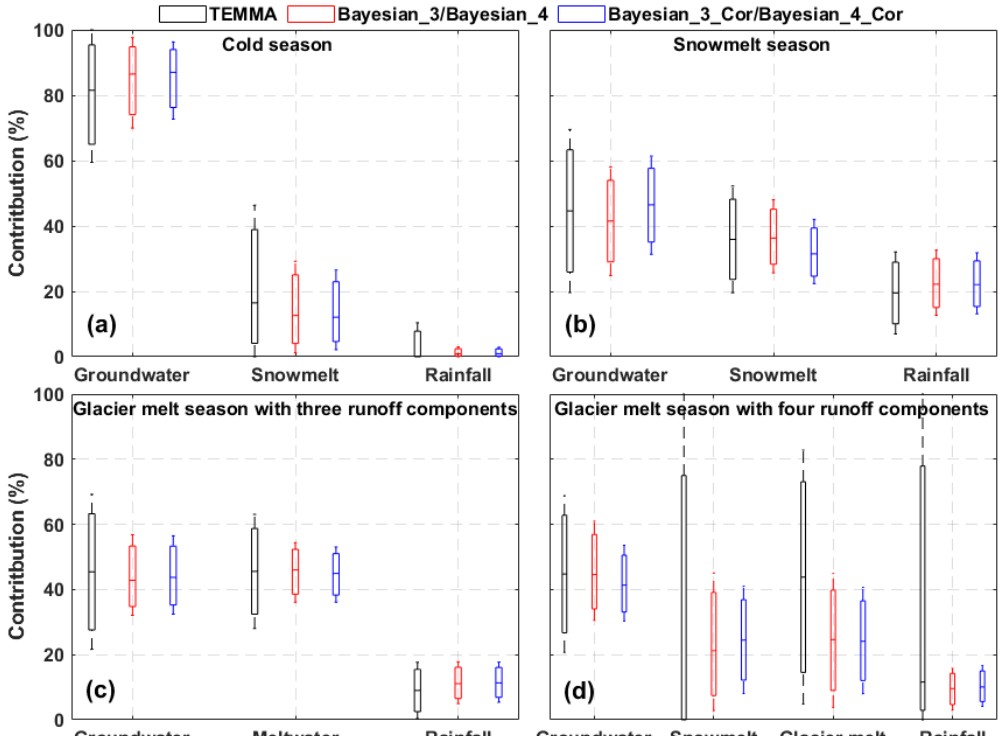

Figure 4. Contributions of runoff components (CRC) to total runoff estimated by different

mixing approaches in three seasons. The Bayesian_3 and Bayesian_3_Cor were applied in the

clod and melt seasons (a-c), and the Bayesian_4 and Bayesian_4_Cor were applied in the

glacier melt season (d). The horizontal lines in the boxes refer to the median contributions,

and whiskers refer to the 95% and 5% percentiles.



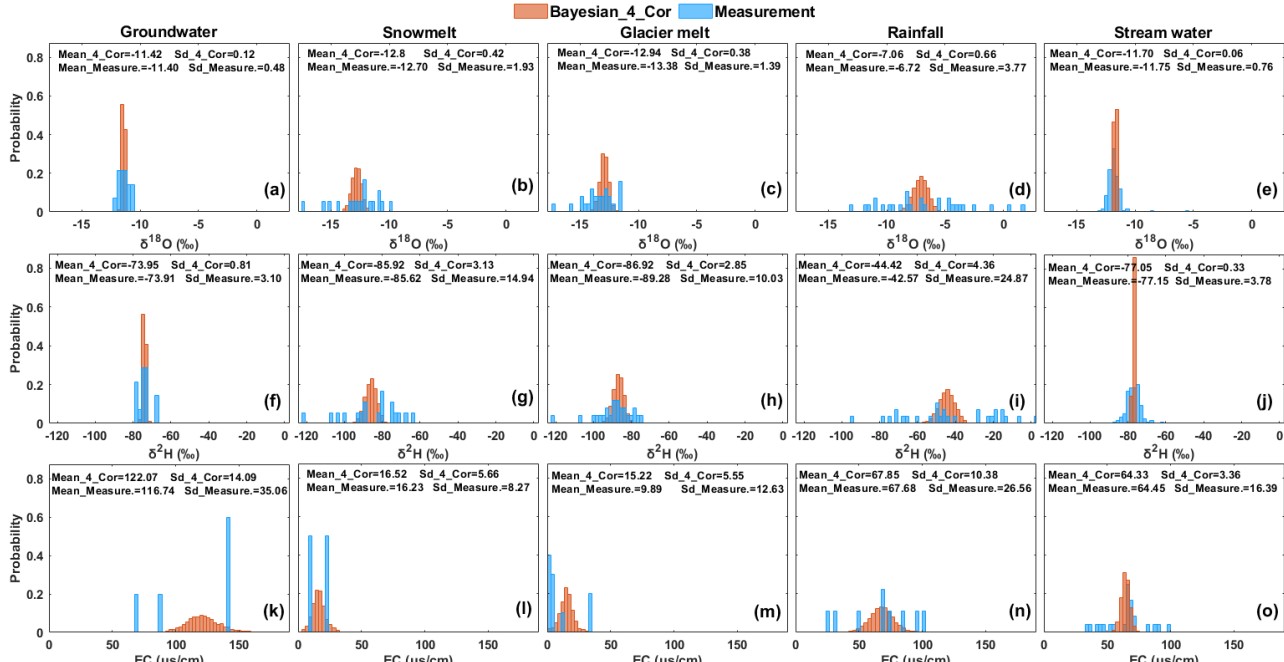

Figure 5. Posterior distributions of water tracer signatures estimated by the Bayesian_4_Cor

in the glacier melt season. Measurement refers to the distributions of water tracer signatures

from the water samples. Row 1: distributions of $\delta^{18}$O; Row 2: distributions of $\delta^2$H; Row 3:

distributions of EC.






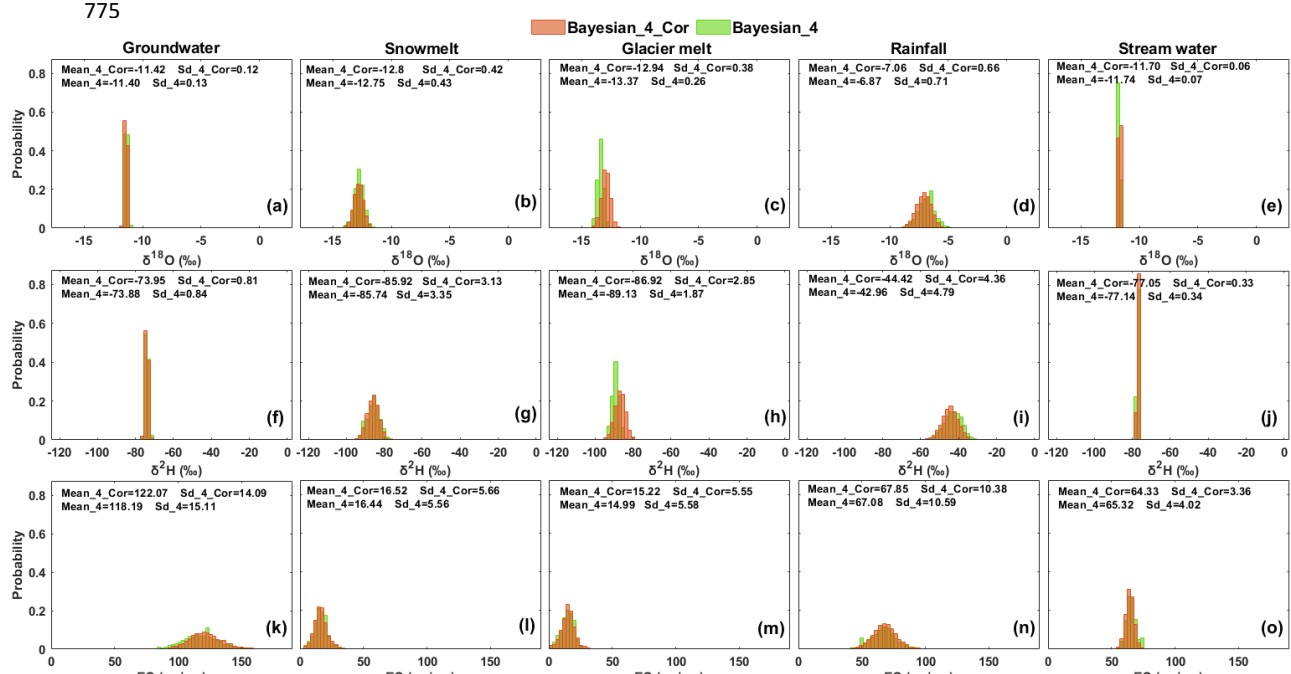

Figure 6. Comparison of the posterior distributions of water tracers estimated by the Bayesian
approaches with (Bayesian_4_Cor) and without (Bayesian_4) considering the correlation
between $\delta^{18}$O and $\delta^{2}$H in the glacier melt season.





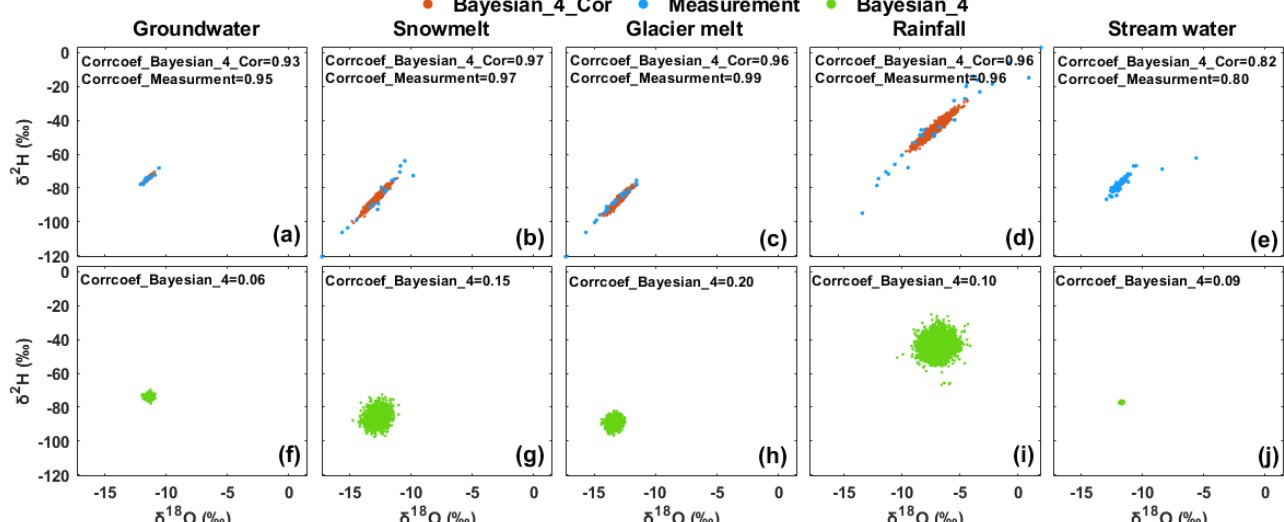

Figure 7. Correlation between posterior $\delta^{18}$O and $\delta^2$H estimated by the Bayesian_4_Cor and

the Bayesian_4 approaches in the glacier melt season.


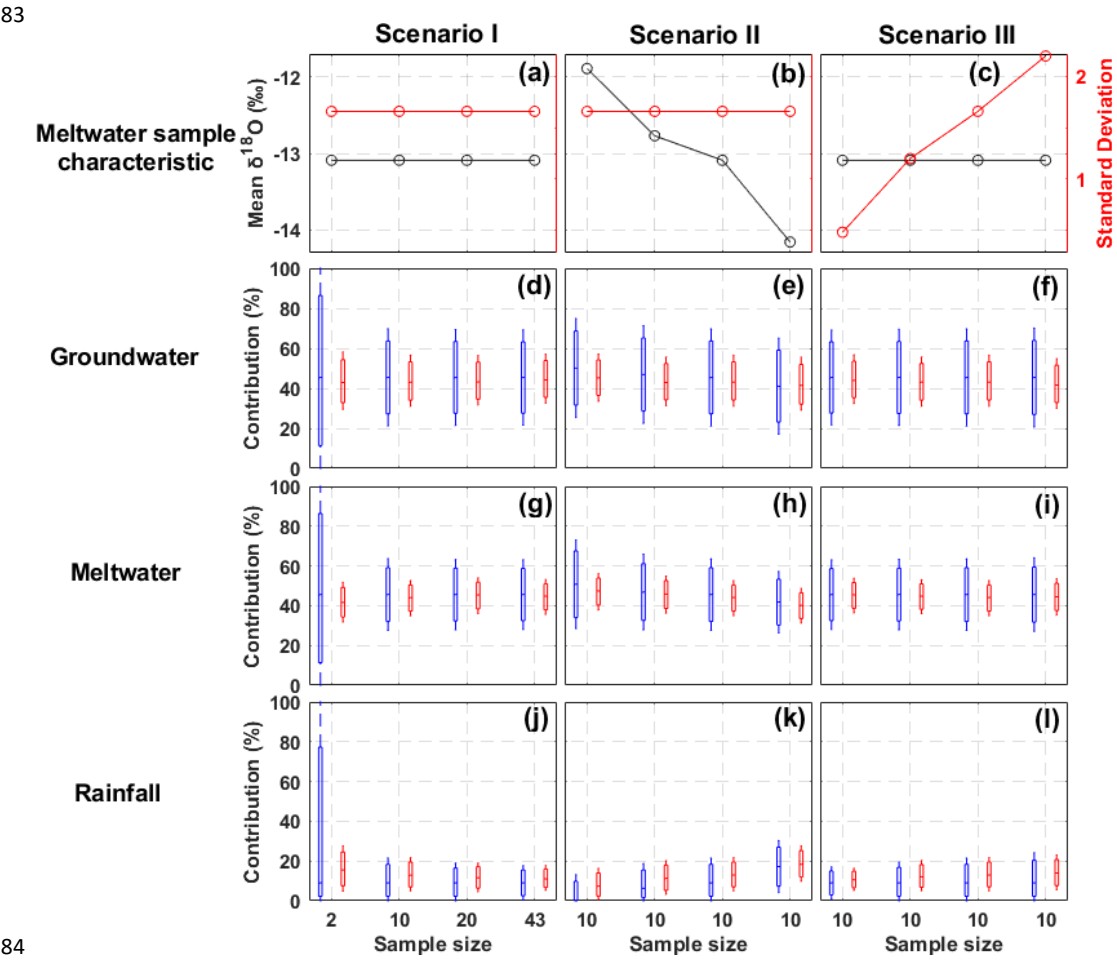


Figure 8. Sensitivity of the estimates for CRC to the sample size (Scenario I), the mean
(Scenario II) and standard deviation (Scenario III) of $\delta^{18}O$ of meltwater in the glacier melt
season. Red boxes show the contributions estimated by the Bayesian_3_Cor, and the blue
boxes refer to the contributions estimated by the TEMMA_3.



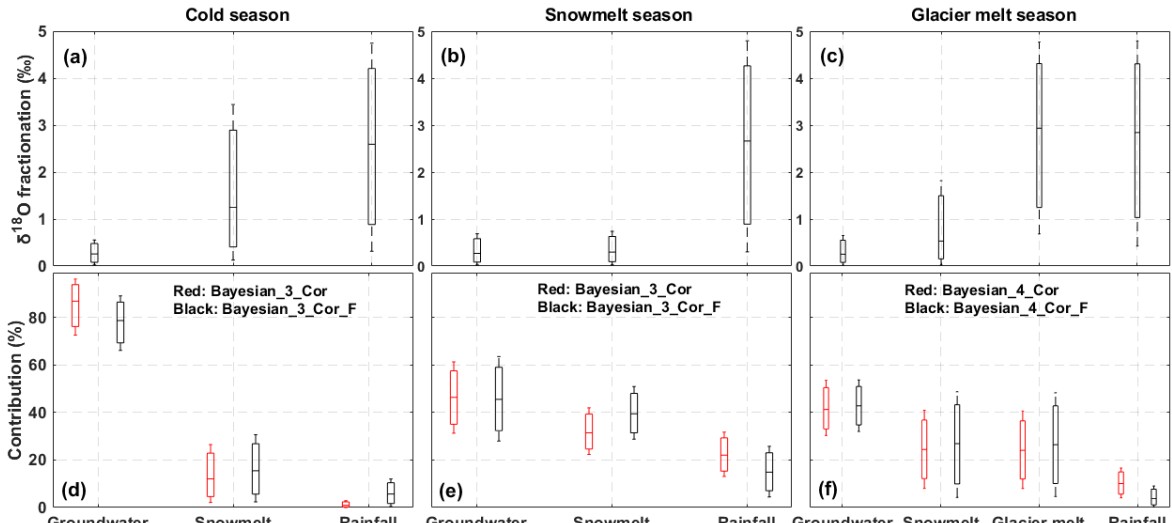

Figure 9. Effects of isotope fractionation on the estimates of CRC in the Bayesian approach
for the three seasons. (a)-(c): Estimated changes in δ¹⁸O of runoff components caused by the
fractionation effect; (d)-(e): Comparison of the CRC estimated by the Bayesian_3_Cor and
the Bayesian_3_Cor_F; (f): Comparison of the CRC estimated by the Bayesian_4_Cor and
the Bayesian_4_Cor_F.





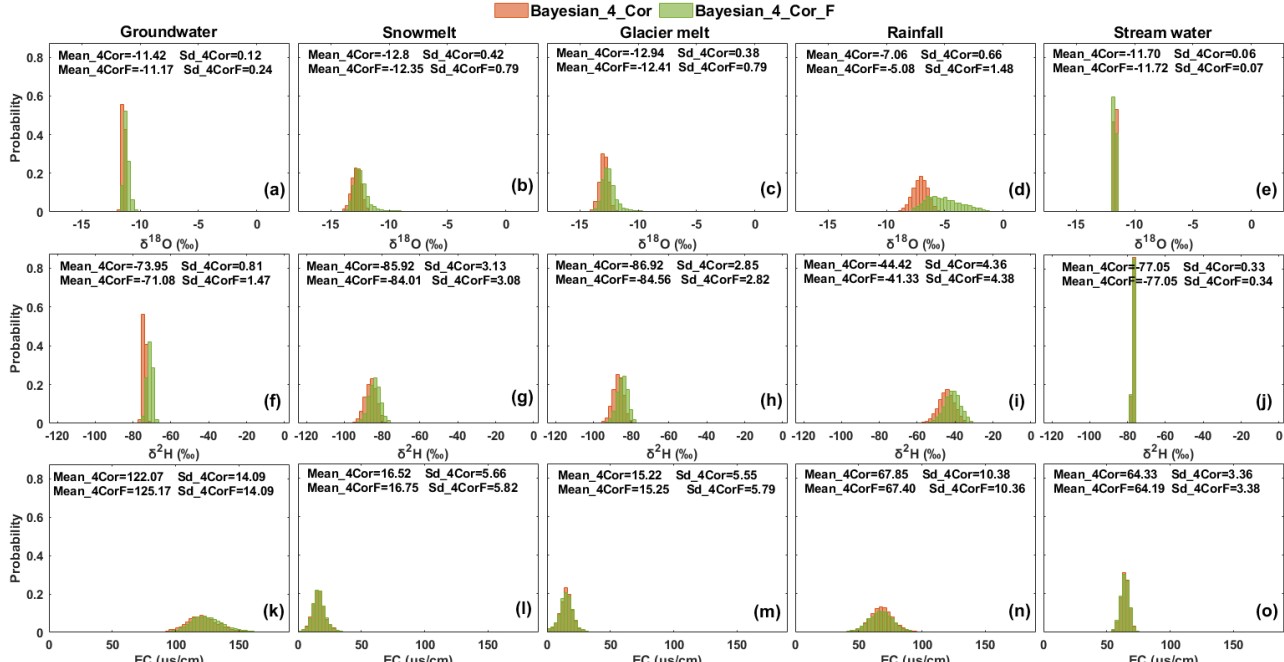

Figure 10. Effects of isotope fractionation on the posterior distributions of tracer signatures of

water sources in the glacier melt season.