# Peer review of "Comparing Bayesian and traditional end-member mixing approaches for hydrograph separation in a glacierized basin"

_Hydrology and Earth System Sciences, 2019_

## Referee Comment (RC1) · Anonymous Referee #1 · 7 Oct 2019

Manuscript: https://doi.org/10.5194/hess-2019-394

Summary: This paper provides an interesting comparison of traditional end-member mixing analysis approaches versus Bayesian statistical approaches for estimating contributions of different runoff components in a glacierized basin in Central Asia. The paper provides an interesting in-depth analysis of the effect of different sources of uncertainty on the Bayesian modeling results. The results clearly highlight that the Bayesian approaches predict more or less the same runoff contributions as the EMMA model when both models have a large sample size, but the Bayesian approach reaches a much smaller uncertainty that is about 50-60% of the EMMA approach. The results

further show that the Bayesian approach is superior to the EMMA approach in situations where sample numbers are low and end members look very similar (e.g. snow and glacier melt signature is similar). The results further show that explicitly considering the correlation between 2H and 18O in the mixing model, further reduces the uncertainty in the results. The paper is well motivated, and the introduction provides a comprehensive overview of the current research on isotope hydrograph separation of runoff components in glacierized basins. The authors explain well the limitations of existing "traditional" approaches such as end-member-mixing-analysis and describe clearly the advantages that Bayesian approaches provide to this problem. My only recommendation would be to add a figure showing the time series of the isotope and EC data and to clarify the "fractionation effect" in the methods and results section. It is currently not clear what this Bayesian modeling scenario encompasses and because of that the section that describes the results of this scenario analysis is confusing. Other than that, the paper, overall, is well written and easy to comprehend. The authors made all relevant code available.

Specific Comments:

Line 146: Please specify what "pure plastic bottles" are? Typically, we state the type (e.g. HDPE or glass) and size of the bottle used to sample water.

Line 108: Please be more specific. What do you mean by "water sampling uncertainty" here? Do you mean the uncertainty associated with having just a few samples?

Line 159: What is the size of the Golubin glacier in the watershed? You mention that glaciers cover about 17% of the watershed. What is the fraction that the Golubin glacier represents in the 17%? What is the streamflow (volume) contribution of the glacier to the entire basin? Is the Golubin glacier representative of the elevation range and snow accumulation of the other glacierized areas in the basin? Did you take grab samples from the other glaciers for comparison? I am a bit concerned that the glacier melt contribution of the Golubin glacier is too small to really make a difference isotopically.

[Figure]

Line 177: Please specify the model and manufacturer of the pH, EC and TDS meter used in this study. Please indicate the precision that this instrument can achieve.

Line 178: How did you determine what constitutes an "abnormal isotopic compositions"? Please describe the method/approach you used.

Line 185: It would be helpful if the authors could add text on how much rainfall and streamflow the Ala-Archa basin typically gets and what the mean annual temperature is. In addition, I would like to suggest providing a graph of the temperature, precipitation and streamflow observed in the Ala-Archa basin between 2012 and 2017 so that the reader can evaluate the interannual variability in the hydro-climate. Since the authors decided to average isotope and EC values across 5 years of observations, this information might help explaining some of the uncertainty in the data.

Line 185: Please add a time series graphs of your isotope and EC, pH and TDS measurements. This graph does not have to be in the main text but could be provided as supplemental material so that the reader can see how the collected data looks like.

Line 250: Please show the histograms of the isotope and EC data. The Bayesian approach assumes that the data is normally distributed, however, based on the data range shown in Figure 3, it looks like that some data might not have been normally distributed? You could report results from a normality test to be sure.

Line 300: It is not quite clear what you mean by "the fractionation effect". Could you be more specific and clarify to the reader when, were this fractionation effect might occur and how it could impact the observed values?

Line 435: The results section on the fractionation effect is confusing. This is mainly because it is not clear what the fractionation effect is and how it is estimated in the sample groups. I would recommend clarifying this in the methods.

Line 463: I would suggest rephrasing to: "The TEMMA estimated similar CRCs for most mixing models but at a larger uncertainty than the Bayesian approaches."

Figure 3: During the glacier melt season the snowmelt end member has a much lower EC value than what was estimated for the cold and snowmelt seasons. Can you explain why the EC is all the sudden so much lower? Since it is most likely not fresh snow that is melting during the glacier melt season, this trend is somewhat surprising.

Minor comments: Line 43: Should be "led" instead of "leaded". Line 114: Use "of" instead of "for the". Line 124: Should be "glaciers cover" instead of "glacier covers" unless you only have one glacier... Line 127: Should be "shows". Line 129: Word missing. Please insert "runoff" after "generates". Line 138: Should be "since the 1960s". Line 158: Should be "was" instead of "were". Line 162: Suggest using "from early March". Line 163: Suggest using "due to" instead of "caused by". Line 168: Please add "meltwater samples". Line 172: "at Helmholtz" Line 183: "split" would be a better word than "distributed". Line 292: please delete "keeping". Line 309: Language! Please rephrase the second part of this sentence. Line 469: Replace "occasionally" with "sporadically". Line 499: Replace "though" with "despite". Line 520: replace "spring points" with "springs". Figure 1: Please remove the underscore for the Rain collector label in the legend.

---

## Referee Comment (RC2) · Anonymous Referee #2 · 8 Oct 2019

[revised manuscript text omitted]
^2 H \end{bmatrix}, \Omega \right) & \text{(6a)} \\[2mm] \boldsymbol{\Omega} \sim Wishart\ (2, \boldsymbol{V}) & \text{(6b)} \\[2mm] \delta^{18}O \sim Normal\ (\mu^{18}O, \lambda^{18}O) & \text{(6c)} \\[2mm] \delta^2 H \sim Normal\ (\mu^2 H, \lambda^2 H) & \text{(6d)} \end{cases}$$

$$\begin{cases} \mu^{18}O \sim Normal\ (\gamma^{18}O, \sigma^{18}O) & \text{(7a)} \\[2mm] \mu^2 H \sim Normal\ (\gamma^2 H, \sigma^2 H) & \text{(7b)} \end{cases}$$

where, $\lambda^{18}$O, $\gamma^{18}$O and $\sigma^{18}$O ($\lambda^2$H, $\gamma^2$H and $\sigma^2$H) are parameters used to describe the normal priors of $\delta^{18}$O and $\mu^{18}$O ($\delta^2$H and $\mu^2$H, see Table 3), which are estimated by likelihood observations (Table 3). $\boldsymbol{V}$ is a 2*2 unit positive-definite matrix, and '2' stands for the degree of freedom in the Wishart prior distribution.

The priors of EC values of runoff components and stream water are assumed as normal distributions (Eq. 8a), with mean $\varepsilon$ and variance $\tau$. Similarly, the spatial variability of the mean EC values of runoff components ($\varepsilon$) are assumed to follow a normal distribution with mean $\theta$ and variance $\omega$ (Eq. 8b). $\tau$, $\theta$ and $\omega$ are parameters estimated by likelihood observations (Table 3).

$$\begin{cases} EC \sim Normal\ (\varepsilon, \tau) & \text{(8a)} \\[2mm] \varepsilon \sim Normal\ (\theta, \omega) & \text{(8b)} \end{cases}$$

$$\begin{cases} \begin{bmatrix} \mu^{18}O \\ \mu^2 H \\ \varepsilon \end{bmatrix}_{stream\ water} = \sum_{i=1}^{N} f_i \cdot \begin{bmatrix} \mu^{18}O \\ \mu^2 H \\ \varepsilon \end{bmatrix}_{runoff\ component\ i} & \text{(9a)} \\[2mm] \boldsymbol{f} \sim Dirichlet(\boldsymbol{\alpha}) & \text{(9b)} \\[2mm] \boldsymbol{\alpha} = \boldsymbol{\rho} + \boldsymbol{\psi} & \text{(9c)} \\[2mm] [\
[revised manuscript text omitted]

---

## Referee Comment (RC3) · Anonymous Referee #3 · 10 Oct 2019

General comments: The study of He and his co-authors presents novel insights into tracer-based hydrograph separation using a comparative approach of evaluating traditional against Bayesian EMMA. In this context, the study aims at filling this important research gap in tracer hydrology both from a methodological and process-oriented point of view. The study shows that the Bayesian approach estimates smaller uncertainties and is less sensitive to sampling uncertainties. The study approach also accounts for isotope fractionation, when using EMMA. Beside only minor comments, I think that the study is mature and presents a concise story line to the readership. The references are with up-to-date and a good use of English can be attributed. After revision of few comments, I can recommend this manuscript for further acceptance in this journal.

[Figure]

Specific comments: Page 6, Line 153: Please use the PALMEX reference (see below) Page 6, Line 175: Please clarify if the measurement precision is the same for both LGR and Picarro instruments, otherwise add this details. Page 6, Line 178: How did you define 'obvious evaporation'? Did you use a deuterium excess threshold? Please insert further details here. Please add also at which EC limit you discarded samples. Page 6, Line 181: Please correct to 'cold season'. Page 8, Line 225: Eqs. 1 -5 hold for 3-components and 2-tracer mixing models. Please provide further information on how you inferred 4 components using 3 tracers. Page 10, Line 293 – 295: Why did you not analyse the snowmelt uncertainty in the snowmelt period? Besides, the sentence is not clear to me: snowmelt is indeed more difficult to sample in the glacier melt season but easier to sample in the snowmelt period. Also its spatio-temporal variability is much higher in that period of time when most of the melting occurs. Page 11, Line 308: Please provide more information on the fractionation effect and how you represented it in your analysis. Page 11, Line 319: It seems that this sentence contradicts with the one in line 326-328. How can glacier melt have high EC if it has low interaction with mineralized surfaces? Please rephrase both parts accordingly. Page 14, Line 379 – 399: This sentence should be moved to the discussion part. Page 15, Line 438: 'In average' Page 16, Line 469: Please clarify. How can samples taken occasionally lead to sharp changes of the isotopic composition? Moreover, randomly taken samples may be part of a strategy to represent tracer variability.

References to add: Gröning, Manfred & Lutz, H.O. & Roller-Lutz, Z. & Kralik, Martin & Gourcy, L. & Pöltenstein, L.. (2012). A simple rain collector preventing water re-evaporation dedicated for $\delta$18O and $\delta$2H analysis of cumulative precipitation samples. Journal of Hydrology. s 448–449. 195–200. 10.1016/j.jhydrol.2012.04.041.

---

## Author Comment (AC1) · 16 Nov 2019

**Reviewer 1:**

**1. Summary: This paper provides an interesting comparison of traditional end-member mixing analysis approaches versus Bayesian statistical approaches for estimating contributions of different runoff components in a glacierized basin in Central Asia. The paper provides an interesting in-depth analysis of the effect of different sources of uncertainty on the Bayesian modeling results. The results clearly highlight that the Bayesian approaches predict more or less the same runoff contributions as the EMMA model when both models have a large sample size, but the Bayesian approach reaches a much smaller uncertainty that is about 50-60% of the EMMA approach. The results further show that the Bayesian approach is superior to the EMMA approach in situations where sample numbers are low and end members look very similar (e.g. snow and glacier melt signature is similar). The results further show that explicitly considering the correlation between $^2$H and $^{18}$O in the mixing model, further reduces the uncertainty in the results. The paper is well motivated, and the introduction provides a comprehensive overview of the current research on isotope hydrograph separation of runoff components in glacierized basins. The authors explain well the limitations of existing "traditional" approaches such as end-member-mixing-analysis and describe clearly the advantages that Bayesian approaches provide to this problem. My only recommendation would be to add a figure showing the time series of the isotope and EC data and to clarify the "fractionation effect" in the methods and results section. It is currently not clear what this Bayesian modeling scenario encompasses and because of that the section that describes the results of this scenario analysis is confusing. Other than that, the paper, overall, is well written and easy to comprehend. The authors made all relevant code available.**

*Reply: Thanks for your positive comments on this paper. We have addressed all your concerns in the revised manuscript. A figure has been added to the supplement to show the time series of water isotope and EC data along with temperature, precipitation and streamflow data. The fractionation effect has been explained in more details, and the related expressions have been refined to reduce confusion.*

**2. Line 146: Please specify what "pure plastic bottles" are? Typically, we state the type (e.g. HDPE or glass) and size of the bottle used to sample water.**

*Reply: We specified the bottles as 50 ml high-density polyethylene (HDPE) bottles in the revised*
*manuscript.*

**3. Line 108: Please be more specific. What do you mean by "water sampling uncertainty"**
**here? Do you mean the uncertainty associated with having just a few samples?**

*Reply: Specified this as "water sampling uncertainty associated with the representativeness of the*
*water samples caused by the limited sample site and sample size". See lines 120-121.*

**4. Line 159: What is the size of the Golubin glacier in the watershed? You mention that**
**glaciers cover about 17% of the watershed. What is the fraction that the Golubin glacier**
**represents in the 17%? What is the streamflow (volume) contribution of the glacier to the**
**entire basin? Is the Golubin glacier representative of the elevation range and snow**
**accumulation of the other glacierized areas in the basin? Did you take grab samples from**
**the other glaciers for comparison? I am a bit concerned that the glacier melt contribution of**
**the Golubin glacier is too small to really make a difference isotopically.**

*Reply: The Golubin glacier has an area of ~5.7 km$^2$ and extends over an elevation range from*
*3232 to 4458 m a.s.l.. The elevation range of the entire glacierized area extends from 3218 to 4857*
*m a.s.l., with about 76% located between 3700 and 4100m a.s.l.. Both the elevation range and the*
*mean elevation (3869 m a.s.l.) of the Golubin glacier are close to those of the entire glacierized*
*area (mean elevation is 3945m a.s.l.). The Golubin glacier represents about 14.4% of the entire*
*glacierized area, while its elevation range covers around 95.6% of the entire glacier range. We*
*only collected meltwater samples from the Golubin glacier, due to the logistic limitations in the*
*remaining glacierized area. Considering the isotopic compositions of snow and glacier meltwater*
*are particularly dependent on the elevation of glacierized area, the sampled meltwater from the*
*Golubin glacier could represent meltwater originated from the primary melting locations in the*
*entire glacierized area. We added these explains in the revised manuscript. See lines 146-150 and*
*625-626.*

**5. Line 177: Please specify the model and manufacturer of the pH, EC and TDS meter used**
**in this study. Please indicate the precision that this instrument can achieve.**

*Reply: Specified as "We used the Hanan Instruments HI-9813 PH EC/TDS portable meter to measure the EC values of water samples, with a measurement precision of 0.1 μs/cm". TDS and pH values of water sample were not recorded. See lines 209-210.*

**6. Line 178: How did you determine what constitutes an "abnormal isotopic compositions"? Please describe the method/approach you used.**

*Reply: We used threshold values to identify abnormal values of $\delta^{18}O$ and EC located far away from the sample clusters. For $\delta^{18}O$, sample values higher than 5‰ were excluded. For EC, sample values higher than 210 μs/cm were excluded. We specified that in the revised manuscript. See lines 214-217.*

**7. Line 185: It would be helpful if the authors could add text on how much rainfall and streamflow the Ala-Archa basin typically gets and what the mean annual temperature is. In addition, I would like to suggest providing a graph of the temperature, precipitation and streamflow observed in the Ala-Archa basin between 2012 and 2017 so that the reader can evaluate the interannual variability in the hydro-climate. Since the authors decided to average isotope and EC values across 5 years of observations, this information might help explaining some of the uncertainty in the data.**

*Reply: A figure for the daily precipitation, temperature and streamflow measured at the basin outlet during 2012-2017 has been added in the supplement (also see the following Figs. S1c-e). Related sentences have been added to describe the hydro-climate data: "The annual mean precipitation and temperature measured at the Baitik meteorological station during 2012-2017 are 538 mm yr$^{-1}$ and 7.2 ℃, respectively. The mean daily streamflow during 2012-2017 is about 6.3 m$^3$/s." The CRC estimated by the mixing approaches refer to the mean contributions in each of the three seasons during the period of 2012-2017. See lines 150-152 and 222.*

[Figure]

*Figure S1. (a)-(b) Tracer signatures of water samples during the sample period of 2012-2017;(c)-(d) Daily precipitation and temperature measured at the Baitik meteorological station in 2012-12017; (e) Daily streamflow measured at the Ala-Archa hydrologic station during 2012-2017.*

**8. Line 185: Please add a time series graphs of your isotope and EC, pH and TDS measurements. This graph does not have to be in the main text but could be provided as supplemental material so that the reader can see how the collected data looks like.**

*Reply: Please, see the above figure. The pH and TDS data were not recorded.*

**9. Line 250: Please show the histograms of the isotope and EC data. The Bayesian approach assumes that the data is normally distributed, however, based on the data range shown in**

**Figure 3, it looks like that some data might not have been normally distributed? You could report results from a normality test to be sure.**

*Reply: Figure 3 only shows the maximum and minimum tracer signatures of each water source. It is not related to the distributions of measured water tracers. The histograms of isotope and EC data in the glacier melt season have been presented in Fig. 5 in the manuscript. A Kolmogorov-Smirnov test has been carried out for both isotope and EC tracers of all water sources. The tracer data of runoff components (i.e., rainfall, snowmelt, groundwater and glacier melt) generally pass the normal distribution test at significance levels of p-values > 0.3, while the tracer data of stream water pass the normal distributions test at significance levels of p-values > 0.1. The EC data of glacier melt pass the normal distribution at a significance level of p-values > 0.06, which can be caused by the low sample size. We thus assume the prior distributions of tracers of runoff components are normal in Eqs. 6-8. The prior distributions of tracers of stream water are first assumed as normal in Eqs. 6a and 8a, and the mean tracer signatures are then calculated by the mixing of tracers of runoff components in Eq. 9. We reported the test results in the revised manuscript. See lines 282-288.*

**10. Line 300: It is not quite clear what you mean by "the fractionation effect". Could you be more specific and clarify to the reader when, where this fractionation effect might occur and how it could impact the observed values?**

*Reply: The water sources for runoff, such as rainfall and meltwater, are subject to evaporation before reaching the basin outlet, especially in summer. However, the isotopic composition of stream water was measured at the basin outlet, and the contributions of runoff components are quantified for the total runoff at the basin outlet. After the long routing path from the sampled sites to the basin outlet, the isotopic compositions of rainfall and meltwater mixing at the basin outlet could be different from those measured at the sampled sites, caused by the evaporation fractionation effect. The isotopic composition of water sources at the sample sites are assumed to be normally distributed in Eqs. 6-7, and the changes in the isotopic compositions of water sources caused by the evaporation fractionation effect are represented by the modification variables $\xi^{18}O$ and $\xi^2H$ in Eq. 10. The evaporation fractionation has no effects on the observed isotopic compositions, but does have one on the quantification of runoff components, which is considered*

*as a source of model uncertainty in the study. We added a more detailed explanation for that in*

*the revised manuscript. See lines 377-393.*

**11. Line 435: The results section on the fractionation effect is confusing. This is mainly**

**because it is not clear what the fractionation effect is and how it is estimated in the sample**

**groups. I would recommend clarifying this in the methods.**

*Reply: Please, see the previous response. We added a more detailed explanation in the method*

*section. See lines 383-393. The quantification of runoff components in two Bayesian scenarios are*

*compared. In the first scenario (using Bayesian_3_Cor and Bayesain_4_Cor), the fractionation*

*effect on isotopic compositions of water sources are ignored, i.e., the isotopic compositions of*

*water sources at the basin outlet are assumed as same as those measured from the sample sites.*

*In the second scenario (using Bayesian_3_Cor_F and Bayesian_4_Cor_F), the evaporation*

*fractionation effect on the isotopic compositions of water sources have been considered. The*

*mixing of water tracers of stream water are represented by Eq. 10. Figure 9 illustrates the effects*

*of fractionation on the quantification of runoff components in all three seasons. The estimated*

*changes in $\delta^{18}O$ of each water source are presented in Figs. 9a-c, and the contributions of runoff*

*components quantified by the two scenarios are compared in Figs. 9d-f.*

**12. Line 463: I would suggest rephrasing to: "The TEMMA estimated similar CRCs for most**

**mixing models but at a larger uncertainty than the Bayesian approaches."**

*Reply: Done. Thanks.*

**13. Figure 3: During the glacier melt season the snowmelt end member has a much lower EC**

**value than what was estimated for the cold and snowmelt seasons. Can you explain why the**

**EC is all the sudden so much lower? Since it is most likely not fresh snow that is melting**

**during the glacier melt season, this trend is somewhat surprising.**

*Reply: In the cold and snowmelt seasons, some snowmelt samples also have EC values as low as*

*those in the glacier melt season (see Fig. 3). The snow samples in the glacier melt season were*

*only collected from the accumulation zone of the glacier, thus resulting in small variability in the*

*EC values. The snowpack in the accumulation zone is accumulated by fresh snow in the snow*

*period (summer type accumulation glacier).This leads to low EC values in the snowmelt samples.*

*We added this discussion in the revised manuscript. See lines 435-440.*

**14. Minor comments: Line 43: Should be "led" instead of "leaded". Line 114: Use "of"**

**instead of "for the". Line 124: Should be "glaciers cover" instead of "glacier covers" unless**

**you only have one glacier: : : Line 127: Should be "shows". Line 129: Word missing. Please**

**insert "runoff" after "generates". Line 138: Should be "since the 1960s". Line 158: Should**

**be "was" instead of "were". Line 162: Suggest using "from early March". Line 163: Suggest**

**using "due to" instead of "caused by". Line 168: Please add "meltwater samples". Line 172:**

**"at Helmholtz" Line 183: "split" would be a better word than "distributed". Line 292: please**

**delete "keeping". Line 309: Language! Please rephrase the second part of this sentence. Line**

**469: Replace "occasionally" with "sporadically". Line 499: Replace "though" with "despite".**

**Line 520: replace "spring points" with "springs". Figure 1: Please remove the underscore**

**for the Rain collector label in the legend.**

*Reply: All done. Thanks.*

---

## Author Comment (AC2) · 16 Nov 2019

**Reviewer 2:**

**1. This is a very interesting and well written manuscript that compares the traditional tracer-based end-member mixing model approach with different versions of a Bayesian mixing model to quantify water sources to runoff in a glacierized catchment in Kyrgyzstan. The findings of this work may have practical implications when applying these approaches to other catchments and are therefore surely interesting to the readers of HESS. The manuscript is logically organized, it is nicely illustrated, the interpretation is well supported by the data, and the discussion is coherent and with relevant and updated references. However, there are some moderate and minor issues that need to be clarified and that I invite the Authors to consider. Please, find these comments, suggestions, and a few corrections in the attached annotated manuscript. I hope they can be useful to the Authors to improve their work.**

*Reply: Thanks a lot for the positive comments. We have addressed all your concerns in this revised manuscript.*

**2. Lines 29 and 143: 'water tracer' to 'tracers' or 'hydrological tracers'; line 38: 'were' to 'was'; line 181: 'clod' to 'cold'; line 418: 'show' to 'shows'; line 490: 'rely' to 'relies'; line 726: Change the sentence into "CV stands for coefficient of variation"; line 734: 'snowmlet' to 'snowmelt'.**

*Reply: All done, thanks.*

**3. Lines 37 and 57: No need to make up a new acronym 'TEMMA'. EMMA is enough, there is no risk to confound it with the other approach.**

*Reply: The used Bayesian method is also a type of end-member mixing approach (EMMA). To avoid the confusion, we used TEMMA to represent the traditional end-member mixing approach.*

**4. Line 70: These are sources of uncertainty that are important in any catchment, not necessarily glacierized catchments. Please, specify why the latter are particularly prone to difficult application of HS (e.g., multiple water sources, high spatio-temporal variability of water sources etc.).**

*Reply: The glacierized catchments are challenging for application of the end-member mixing approaches because of the following reasons: (1) The catchment elevation generally extends over a large range, leading to strong spatial variability in climate forcing (precipitation and*

*temperature) and the tracer signatures of water sources; (2) The number of end-member water*

*sources for runoff is high, additionally including snow and glacier meltwater; (3) Water sampling*

*in high-elevation glacierized catchment is difficult due to the logistical limitations, resulting in*

*small sample sizes for the application of (T)EMMA. We specified these in the revised manuscript.*

*See lines 67-73.*

**5. Line 77: But only the statistical uncertainty! Please, specify.**

*Reply: Specified as the "statistical uncertainty" in this manuscript.*

**6. Lines 83-87: This two issues are important but not very clearly explain. Please, clarify.**

*Reply: We refined these sentences as follows: These include (1) inappropriate estimation of the*

*variability of tracer signatures of water sources when only few water samples are available. The*

*used Sd values of the measured tracer signatures likely fail to represent the variability of water*

*tracer signature of individual water source across the basin, due to the small water sample sizes;*

*(2) The correlation of tracer signatures and runoff components are inevitably ignored, due to the*

*assumption of independence of the multiple uncertainty sources. The correlation between $\delta^{18}O$ and*

*$\delta^2H$ of each water source, as well as the interaction between runoff components could provide*

*additional constraints on the uncertainty in the quantification of runoff components, which*

*however are typically ignored in the Gaussian error propagation technique. See lines 88-97.*

**7. Line 93: In this paragraph it's important, in my opinion, to add a description on how**

**uncertainty is treated in the Bayesian approach. This is particularly important for the**

**research question #2.**

*Reply: In the Bayesian approach, both the statistical and model uncertainty are represented by*

*the posterior distributions of parameters. The parameter uncertainty is estimated based on*

*likelihood observations using a Markov Chain Monte Carlo procedure. This explanation has been*

*added in the revised manuscript. See lines 106-109.*

**8. Line 109: How do Bayesian mixing models estimate the isotopic fractionation? I suggest**

**to add a sentence here.**

*Reply: Modified as "Benefiting from the prior assumptions for changes in isotope signatures*

*during the mixing process, the Bayesian approach bears the potential to estimate the fractionation*

*effect on isotopic signatures, which however, has not been investigated either." See lines 122-124.*

**9. Line 113: In the two research questions outlined here it is not adequately stressed/explained why a glacierized catchment has been chosen for addressing these questions. Indeed, they can be applied to any catchment. Please, specify this.**

*Reply: We added a more detailed explanation here: "In Central Asia, glacierized catchments provide important fresh water supply for downstream cities and irrigated agriculture. Quantifying the contributions of multiple runoff components to total runoff is important for understanding the dynamics of water resource availability at the regional scale. However, uncertainty in the quantification of runoff components in the glacierized catchments are particularly large because of the following reasons: (1) The catchment elevation generally extends over a large range, leading to strong spatial variability in climate forces (precipitation and temperature) and the tracer signatures of water sources; (2) The number of end-member water sources is large, additionally including snow and glacier meltwater; (3)Water sampling in high-elevation glacierized catchments is difficult due to the logistical limitations, resulting in small sample sizes to represent the tracer signatures of water sources." See lines 127-131.*

**10. Line 143: As we know, EC is not as conservative as tracers. However, due to its easy use it has been often applied in catchment studies. Please, include a short discussion on the possible issue related to the lack of conservative behaviour (e.g., not so relevant at the catchmen scale, or at the runoff event scale etc.)**

*Reply: We added related discussion on this issue as follows: "EC data has been widely used for hydrograph separation, due to its easy use and quick measurement. While EC is not a conservative tracer, this may have only small effects on the application of hydrograph separation at the catchment scale." See lines 210-213.*

**11. Line 175: Any procedure to minimize memory effect (carry over effect) was performed?**

*Reply: Added: "A regular re-calibration procedure has been carried out for the isotope analysis." See line 206.*

**12. Line 176: First time it's mentioned...define electrical conductivity.**

*Reply: Defined in line 61.*

**13. Line 177: Can you quantify the term "abnormal"?**

*Reply: We used threshold values to identify abnormal values of $\delta^{18}O$ and EC located far away from the sample clusters. For $\delta^{18}O$, sample values higher than 5‰ were excluded. For EC, sample*

*values higher than 210 µs/cm were excluded. We specified that in the revised manuscript. See lines*

*214-217.*

**14. Line 227: It's not clear to me how 4-component HS can be performed using two tracers**

**only. Indeed, due to the collinearity of 18oxygen and deuterium, these two tracers cannot be**

**treated independently. So, how are mixing approaches TEMMA4, Bay4 and Bay4cor defined?**

**Please, this parts need to be extremely clear to the readers.**

*Reply: Yes, the values of $\delta^{18}O$ and $\delta^2H$ are typically correlated for each water source. However,*

*the coefficients representing the correlation between $\delta^{18}O$ and $\delta^2H$ vary among the water sources*

*in glacierized catchment (see Fig. 2), thus providing a basis for the TEMMA_4 to quantify four*

*runoff components. When quantifying four runoff components using three tracers, four*

*conservative equations for $\delta^{18}O$, $\delta^2H$, EC, and water volume are used (similar to Eq.1). The*

*contributions of runoff components ($f$), as well as the partial derivatives used to calculate the*

*uncertainty are solved from the four conservative equations using Matlab. However, the solutions*

*are too lengthy to show in the text. As expected, results in Table 4 show that the TEMMA_4 failed*

*to distinguish snowmelt and glacier melt runoff, due to the similar tracer signatures of these two*

*runoff components, but succeeded in quantifying the contributions of rainfall and groundwater.*

*The Bayesian_4 and Bayesian_4_Cor estimated the contributions of four runoff components based*

*on the prior distributions of $\delta^{18}O$, $\delta^2H$ and EC. The correlation between $\delta^{18}O$ and $\delta^2H$ is ignored*

*in Bayesian_4. We used independent prior distributions for $\delta^{18}O$ and $\delta^2H$ of each water source. In*

*Bayesian_4_Cor, parameters describing the correlation between $\delta^{18}O$ and $\delta^2H$ of each water*

*source were estimated by likelihood observations of the corresponding water source, which also*

*vary among the water sources, thus providing a basis for the quantification of four runoff*

*components using four mixing equations of tracer signatures (similar to Eq.9). The four-*

*components approaches are developed in our study to investigate the following two questions: (1)*

*Is the TEMMA able to quantify four runoff components just using $\delta^{18}O$, $\delta^2H$, and EC? (2) Does the*

*correlation between $\delta^{18}O$ and $\delta^2H$ help to reduce the uncertainty in the quantification of runoff*

*components? We added these explains in the revised manuscript. See lines 267-274 and 337-346.*

**15. Line 288: The three scenarios are not immediately clear. Does the mean refer to the**

**spatial value or the temporal value, or the spatial-temporal value? The same question applies**

**to sd. Then, different compared to what? Please, specify.**

*Reply: Meltwater sampling in glacierized catchments is typically difficult due to the logistic*

*limitations. Thus, a small number of samples from a few sites are usually used for hydrograph*

*separation. The uncertainty in the representativeness of meltwater samples implies an additional*

*uncertainty source for quantification of runoff components. To investigate the effects of this type*

*of sampling uncertainty, we set up three virtual sampling scenarios. Scenario I is used to evaluate*

*the effects of meltwater sample size, in which four groups of meltwater sample are tested. The four*

*sample groups have the same mean value and Sd of $\delta^{18}O$ or EC, but different sample sizes. Mean*

*and Sd values of $\delta^{18}O$ or EC are calculated for all used meltwater samples in each group, referring*

*to the spatio-temporal variability (same in the following two scenarios). Scenario II is used to*

*evaluate the effects of sampled mean value of $\delta^{18}O$ (or EC) of meltwater. The four sample groups*

*have the same sample size and Sd, but different mean values. Scenario III is used to investigate*

*the effects of Sd values of sampled $\delta^{18}O$ (or EC). The four sample groups have the same sample*

*size and mean tracer signature, but different Sd values. See lines 348-362.*

**16. Line 330: This is not clearly understandable from the table. Consider replacing it with a**

**boxplot.**

*Reply: Done. See Figs. 3 in the revised manuscript.*

**17. Line 346: So, do the bars represent the spatio-temporal standard deviation?**

*Reply: The bars just represent the minimum and maximum values of each tracer signature.*

**18. Line 356: This sentence is not clear. Please, specify.**

*Reply: Modified as "Tracer signatures of rainfall are assumed as the same as the tracer signatures*

*of precipitation samples in all the three seasons". See line 227.*

**19. Line 466: This holds true for this specific study and perhaps for other catchments (not**

**only glaicerized) but not necessarily for all. This should be noted in the discussion.**

*Reply: Modified as "Sd values are likely overestimated in this study due to small sample sizes, and*

*thus insufficiently representing the variability of the tracer signatures of the corresponding water*

*sources across the basin." See lines 564-566.*

**20. Line 469: Sampling occasionally not necessarily lead to sharp changes! Please, explain.**

*Reply: Modified as "Due to the limited accessibility of the sampled sites caused by snow cover,*

*the samples of meltwater and groundwater are often collected sporadically. The small sample size*

*and strong variability in sampled tracers likely lead to a large Sd value." See lines 566-568.*

**21. Table 1: This table is quite long and dense. Please, consider replacing it with box-plots.**

*Reply: This table has been split into three sub-tables. Boxplots have been added to present the*

*variability of tracer signatures.*

**22. Table 4: Perhaps reporting the mean and the SD is clearer than reporting the mean and**

**the range. Please, consider this possible change.**

*Reply: The ranges of minimum and maximum contributions are used to represent the uncertainty*

*ranges. Sd values have been added in the table. See Table 4.*

---

## Author Comment (AC3) · 16 Nov 2019

**1. General comments: The study of He and his co-authors presents novel insights into tracer-based hydrograph separation using a comparative approach of evaluating traditional against Bayesian EMMA. In this context, the study aims at filling this important research gap in tracer hydrology both from a methodological and process-oriented point of view. The study shows that the Bayesian approach estimates smaller uncertainties and is less sensitive to sampling uncertainties. The study approach also accounts for isotope fractionation, when using EMMA. Beside only minor comments, I think that the study is mature and presents a concise story line to the readership. The references are with up-to-date and a good use of English can be attributed. After revision of few comments, I can recommend this manuscript for further acceptance in this journal.**

*Reply: Thanks a lot for the positive comments. We have addressed all your concerns in this revised manuscript.*

**2. Page 6, Line 153: Please use the PALMEX reference (see below).**

*Reply: Done. Thanks.*

**3. Page 6, Line 175: Please clarify if the measurement precision is the same for both LGR and Picarro instruments, otherwise add this details.**

*Reply: Both measurement precisions of $\delta^{18}O$ and $\delta^2H$ are $\pm0.25$ ‰ and $\pm0.4$ ‰, respectively. Specified in the revised manuscript. See line 207.*

**4. Page 6, Line 178: How did you define 'obvious evaporation'? Did you use a deuterium excess threshold? Please insert further details here. Please add also at which EC limit you discarded samples.**

*Reply: We used threshold values to identify abnormal values of $\delta^{18}O$ and EC located far away from the sample clusters. For $\delta^{18}O$, sample values higher than 5‰ were excluded. For EC, sample values higher than 210 µs/cm were excluded. We specified that in the revised manuscript. See lines 214-217.*

**5. Page 6, Line 181: Please correct to 'cold season'; Page 15, Line 438: 'In average'.**

*Reply: Done. Thanks.*

**6. Page 8, Line 225: Eqs. 1 -5 hold for 3-components and 2-tracer mixing models. Please provide further information on how you inferred 4 components using 3 tracers.**

*Reply: When quantifying four runoff components using three tracers, four conservative equations for $\delta^{18}O$, $\delta^2H$, EC, and water volume are used (similar to Eq.1). The values of $\delta^{18}O$ and $\delta^2H$ are typically correlated for each runoff component. However, the coefficients representing the correlation between $\delta^{18}O$ and $\delta^2H$ vary among the runoff components in glacierized catchment (see Figure 2), thus providing a basis for the TEMMA_4 to quantify four runoff components using four conservation equations. The contributions of runoff components ($f_i$) as well as the partial derivatives used to calculate the uncertainty are solved from the four conservative equations using Matlab. However, the solutions are too lengthy to show in the text. We specified these in the revised manuscript. See lines 267-274.*

**7. Page 10, Line 293 – 295: Why did you not analyze the snowmelt uncertainty in the snowmelt period? Besides, the sentence is not clear to me: snowmelt is indeed more difficult to sample in the glacier melt season but easier to sample in the snowmelt period. Also its spatio-temporal variability is much higher in that period of time when most of the melting occurs.**

*Reply: We investigated the effects of sampling uncertainty only in the glacier melt season because of the following two reasons: (1) Runoff in the glacier melt season contributes the largest part to annual runoff in our study basin. Accurate quantification of each runoff component in this season is extremely important for the understanding of dynamics of water availability in the study area. (2) In this season more meltwater samples are available (15 snowmelt samples and 23 glacier melt samples) than in the snowmelt season (only 15 snowmelt, Table 1), thus providing a good observation data basis for the investigation experiment. Snowmelt sampling in the snowmelt season in the study basin is also difficult due to the heavy snow accumulation in March to April and the spring flood in May to June. However, we believe the effects of snowmelt sampling uncertainty on the end-member mixing approaches in the snowmelt season should be similar to those of meltwater sampling in the glacier melt season. We explained this issue in the revised manuscript. See lines 368-375.*

**8. Page 11, Line 308: Please provide more information on the fractionation effect and how you represented it in your analysis.**

*Reply: The water sources for runoff, such as rainfall and meltwater, are subject to evaporation before reaching the basin outlet, especially in summer. However, the isotopic composition of stream water was measured at the basin outlet, and the contributions of runoff components are*

*quantified for the total runoff at the basin outlet. After the long routing path from the sampled sites to the basin outlet, the isotopic compositions of rainfall and meltwater mixing at the basin outlet could be different from those measured at the sampled sites, caused by the evaporation fractionation effect. The isotopic composition of water sources at the sample sites are assumed to be normally distributed in Eqs. 6-7, and the changes in the isotopic compositions of water sources caused by the evaporation fractionation effect are represented by the modification variables $\xi^{18}O$ and $\xi^{2}H$ in Eq. 10. Parameters describing the prior distributions of isotopic compositions at the sample sites in Eqs. 6-7 are estimated by the likelihood observations of isotope signatures of water samples. The modification variables $\xi^{18}O$ and $\xi^{2}H$ are estimated by the likelihood observations of isotope signatures of stream water. The fractionation effect on the estimated CRC is quantified by comparing two Bayesian scenarios. In the first scenario (using Bayesian_3_Cor and Bayesain_4_Cor), the isotopic compositions of water sources at the basin outlet are assumed the same as those measured from the sample sites even though the water sources have suffered evaporation before reaching the basin outlet (using Eqs. 6-9). In the second scenario (using Bayesian_3_Cor_F and Bayesian_4_Cor_F), the evaporation fractionation effect on the isotopic compositions of water sources is considered, and the mixing of water tracers of stream water is represented by Eq.10. We added these explains in the revised manuscript. See lines 377-393. Figure 9 illustrates the effects of fractionation on the quantification of runoff components in all three seasons. The estimated changes in $\delta^{18}O$ of each water source are presented in Figs. 9a-c, and the contributions of runoff components quantified by the two scenarios are compared in Figs. 9d-f*

**9. .Page 11, Line 319: It seems that this sentence contradicts with the one in line 326-328. How can glacier melt have high EC if it has low interaction with mineralized surfaces? Please rephrase both parts accordingly.**

*Reply: Line 319 has been modified as: "Among the water sources, snowmelt and glacier melt tend to have the lowest EC values, due to low interaction with mineral surface." Lines 326-328 have been rephrased as: "The highest CV value of EC for glacier melt indicates large variability in the glacier melt samples. This is because the glacier melt water samples were collected from a rather clean location (EC value is only 1.5 µs/cm) and a relatively dusty location (EC value is 33.4 µs/cm)." See lines 411-412 and 425-428.*

**10. Page 14, Line 379 –381: This sentence should be moved to the discussion part.**

*Reply: Modified as: "The TEMMA_3 estimated the largest uncertainty ranges and Sd values for CRC in all the three seasons, followed by the Bayesian_3."*

**11. Page 16, Line 469: Please clarify. How can samples taken occasionally lead to sharp changes of the isotopic composition? Moreover, randomly taken samples may be part of a strategy to represent tracer variability.**

*Reply: Modified as "Due to the limited accessibility of the sample sites caused by snow cover, the water samples of meltwater and groundwater are often collected sporadically. The small sample size and strong variability in sampled tracers likely lead to a large Sd value." See lines 566-568.*